EMBO
Molecular Medicine

# Brown remodeling of white adipose tissue protects against abdominal aortic aneurysm via batokine FSTL1

Chunling Huang [1,2,7], Yuna Huang [3,7], Boshui Huang [4,5,7], Lei Yao[1,2,7], Zenghui Zhang[1,4,5], Luoxiao Dong[1,2], Chang Guan[6], Junping Li[1,4,5], Zhaoqi Huang[1,4,5], Sixu Chen[1,4,5], Yuan Jiang[1,4], Yuling Zhang[4,5], Jingfeng Wang [4,5✉], Yangxin Chen [4,5✉] & Zhaoyu Liu [1,2✉]

## Abstract

**Abdominal aortic aneurysm (AAA) is a life-threatening vascular disease without effective medical therapies. Emerging evidence have suggested a crosstalk between adipose tissue and vascular cells. Besides, brown adipose tissue is considered beneficial for cardiovascular health. Nevertheless, whether brown remodeling of white adipose tissue would protect against AAA remains unclear. Here, we showed that patients with AAA had a decreased browning level of adipose tissue, and induction of adipose tissue browning significantly reduced AAA incidence and attenuated AAA development in mice. Using LC-MS/MS and proteomic analysis, we further identified Follistatin-like 1 (FSTL1) as a novel vessel-protective adipokine secreted by browning adipocytes. Mechanistically, FSTL1 inhibited VSMC apoptosis through DIP2A/AKT signaling. Furthermore, we demonstrated that adipocyte-specific deficiency of FSTL1 abrogated the protective effect of browning induction. Moreover, supplementation of FSTL1 either systemically or patched into hydrogel placing around the abdominal aorta markedly limited aortic dilation and AAA progression. Our data suggest a protective role of adipose tissue browning and batokine FSTL1 in the development of AAA, which may represent a novel intervention strategy for AAA.**

**Keywords** Abdominal Aortic Aneurysm; Adipose Tissue; Browning; FSTL1
**Subject Categories** Cardiovascular System; Metabolism

## Introduction

Abdominal aortic aneurysm (AAA) is a life-threatening vascular disease characterized by abnormal focal dilation of the abdominal aorta, which is mostly asymptomatic but can lead to sudden death due to aortic rupture (Kent, 2014). Current management of AAA is limited to surgical repair (Golledge, 2019), whereas clinical trials indicated that early elective surgical intervention for people with small AAAs does not reduce mortality (United Kingdom Small Aneurysm Trial et al, 2002). Most small AAAs continue to expand, and the risk of rupture increases with increasing aneurysm diameter (1998). Despite extensive research, there are no approved pharmacological therapies that can effectively prevent or decelerate AAA progression.

Adipose tissue is an important organ not merely for lipid storage and energy homeostasis but also an important source of bioactive molecules that regulate cardiovascular physiology (Polkinghorne et al, 2024). Several clinical and animal studies have suggested that excessive adipose tissue, mostly white adipose tissue accumulation, heightens the risk of developing AAA (Golledge et al, 2007; Kugo et al, 2019; Stackelberg et al, 2013). However, surgical removal of the adipose tissue is not clinically feasible for AAA intervention. Therefore, to remodel those adipose tissues to reduce their adverse effects while enhancing their beneficial effects may offer a better strategy. Recently, a large clinical study has revealed a potential role of brown adipose tissue in promoting cardiometabolic health (Becher et al, 2021). In mouse studies, brown adipose tissue has been found to protect against pathological cardiac remodeling and decrease myocardial damage (Lin et al, 2022; Marti-Pamies et al, 2023; Pinckard et al, 2021; Ruan et al, 2018). Besides, brown adipose tissue and brown remodeling of perivascular adipose tissue have also been demonstrated to protect from vascular dysfunction such as atherosclerosis (Adachi et al, 2022; Berbee et al, 2015; Shi et al, 2022). However, whether brown remodeling of white adipose tissue would protect against AAA development is largely unknown.

[1]Medical Research Center, Sun Yat-Sen Memorial Hospital, Sun Yat-Sen University, Guangzhou, China. [2]Guangdong Provincial Key Laboratory of Malignant Tumor Epigenetics and Gene Regulation, Sun Yat-Sen Memorial Hospital, Sun Yat-Sen University, Guangzhou, China. [3]Department of Cardiovascular Ultrasound, Zhongnan Hospital of Wuhan University, Wuhan University, Wuhan, China. [4]Department of Cardiology, Guangdong Provincial Key Laboratory of Arrhythmia and Electrophysiology, Sun Yat-Sen Memorial Hospital, Sun Yat-Sen University, Guangzhou, China. [5]Guangzhou Key Laboratory of Molecular and Mechanisms of Major Cardiovascular Disease, Sun Yat-Sen Memorial Hospital, Sun Yat-Sen University, Guangzhou, China. [6]Department of Cardiology, the First Affiliated Hospital of Guangzhou Medical University, Guangzhou, China. [7]These authors contributed equally: Chunling Huang, Yuna Huang, Boshui Huang, Lei Yao.✉E-mail: wjingf@mail.sysu.edu.cn; chenyx39@mail.sysu.edu.cn; liuzhy98@mail.sysu.edu.cn

The loss of vascular smooth muscle cells (VSMC) is one of the major features in the pathogenesis of AAA (Lu et al, 2020). The reduced VSMC number in the aortic wall impairs their ability to produce connective tissue and repair elastin breaks, which leads to a weakened vessel and subsequent dilatation of the aorta (Lu et al, 2021). The apoptosis of VSMCs is a major cause of VSMC depletion in AAA development (Rowe et al, 2000; Thompson et al, 1997). Multiple studies have indicated an interplay between adipose tissue and the vascular system, in an endocrine or paracrine manner, through secreting adipokines and other factors (Akoumianakis et al, 2017; Wang et al, 2009; Zhang et al, 2018). Nevertheless, whether browning adipocytes secrete VSMC-protective factors remains to be elucidated.

In this study, we investigated the functional role of adipose tissue browning in AAA development. We found that browning induction significantly enhanced vessel integrity and attenuated AAA development in mice. In addition, browning adipocytes secreted Follistatin-like 1(FSTL1) and inhibited VSMC apoptosis. Supplementation of recombinant FSTL1 markedly inhibited AAA progression, indicating its therapeutic potential for AAA intervention.

# Results

## Decreased browning level correlates with AAA development in human

To investigate the association between the browning level of adipose tissue and AAA development, we first analyzed the expression of classic browning marker UCP-1 (uncoupling protein 1) in perivascular adipose tissue (PVAT) in patients with/without AAA. Compared to non-AAA patients, there was a significant reduction in UCP-1 levels in adipocytes of PVAT from patients with AAA, as determined by immunofluorescence staining (Fig. 1A,B). This decrease in UCP-1 expression was further confirmed at both the mRNA and protein levels by quantitative real-time PCR and western blot analysis, respectively (Fig. 1C, D). These data suggested a decreased browning level with AAA development. To further confirm this association, we carried out paired comparison analysis of adipose tissue surrounding the dilated abdominal aorta and non-dilated aortic neck in AAA patients using data from the Gene Expression Omnibus (GEO) database (GSE119717) (Piacentini et al, 2019). The results indicated that browning-associated genes such as PGC1α, COX7A1, COX7A2, and COX4I1 were significantly decreased in dilated PVAT compared to non-dilated PVAT (Fig. 1E–H). These data suggested that decreased browning level correlates with aortic dilation and possibly AAA progression.

## Brown remodeling of white adipose tissue protects against AAA in vivo

To determine the role of brown remodeling of white adipose tissue in AAA development, we administered CL316,243, a drug widely used for browning induction, or saline control to 5-month-old ApoE$^{-/-}$ mice and then constructed an angiotensin II (Ang II)-induced AAA model (Fig. 1I). After 9 days of administration of CL316,243, brown remodeling was successfully induced in both epididymal white adipose tissues (eWAT) and subcutaneous white adipose tissues (sWAT), as indicated by a more brownish gross appearance (Fig. EV1A), more adipocytes with small lipid droplets

in a multilocular pattern (Fig. EV1B), and a much higher Ucp-1 expression (Fig. EV1C). In addition, perivascular adipose tissue around the abdominal aorta also showed a higher Ucp-1 expression as indicated by immunohistochemical staining (Fig. EV1D). However, mice's body weight and blood glucose levels were not changed (Fig. EV2A,B). Further analysis of serum lipids revealed a significant reduction in triglyceride levels but unchanged cholesterol levels (Fig. EV2C,D). Notably, CL316,243 administration significantly reduced AAA incidence (76.5% vs 16.7%, Fig. 1J) and decreased the maximal diameter of the abdominal aorta compared with saline control after Ang II infusion, as indicated by ex vivo analyses and ultrasound imaging (Fig. 1K–M). Histological analysis revealed that CL316,243 administration reduced elastin fragmentation in the abdominal aortas, as determined by Elastica van Gieson staining (Fig. 1N,O). Collectively, these results suggested that browning remodeling of white adipose tissue protects against AAA development in vivo. As hypertension has been suggested as one of the most important risk factors for AAA (Iribarren et al, 2007), we further examined the blood pressure in CL316,243-treated mice. Nevertheless, CL316,243 administration did not significantly change systolic or diastolic blood pressures with or without Ang II infusion, as compared to saline control (Fig. EV3A,B), suggesting that the protective role of CL316,243 during AAA is independent of blood pressure regulation.

## Brown remodeling of white adipose tissue inhibits VSMC apoptosis in AAA

To gain insight into the mechanisms underlying the protective effects of adipose tissue browning on AAA, we performed a high-throughput RNA-sequencing of aortic tissues from mice after saline or CL316,243 treatment and Ang II modeling. A total of 911 genes were differentially expressed in mice after CL316,243 treatment, with 355 upregulated and 556 downregulated (Fig. 2A). Gene Ontology analysis revealed that the upregulated genes were significantly enriched in cell matrix adhesion and negative regulation of apoptotic process (Fig. 2B). As smooth muscle cells (SMCs) are the major component of the arterial wall and their apoptosis contributes to degeneration of aortic extracellular matrix and subsequent AAA formation and progression (Lu et al, 2020; Quintana et al, 2019a), we thus went on to investigate the content of SMCs. The results showed that Ang II treatment significantly decreased SMC content in the aortic wall, whereas CL316,243 administration significantly reversed this decrease as revealed by immunofluorescence staining of alpha-smooth muscle actin (SMA) (Fig. 2C,D). However, the markers for VSMC phenotypic switch, such as krüppel-like factor 4 (KLF4) and osteopontin (OPN), were unchanged (Fig. EV4). In addition, the apoptosis marker cleaved-Caspase3 was significantly decreased (Fig. 2E,F). Co-staining of TUNEL (terminal deoxynucleotidyl transferase dUTP nick end labeling) and SMA further indicated that apoptosis of VSMCs was significantly decreased by CL316,243 administration (Fig. 2G,H). All these results suggested that brown remodeling of white adipose tissue inhibits VSMC apoptosis in AAA.

## Secretome of browning adipocytes inhibits VSMC apoptosis

Next, we investigated whether browning agent CL316,243 has a direct effect on VSMC apoptosis. Annexin V/PI staining combined

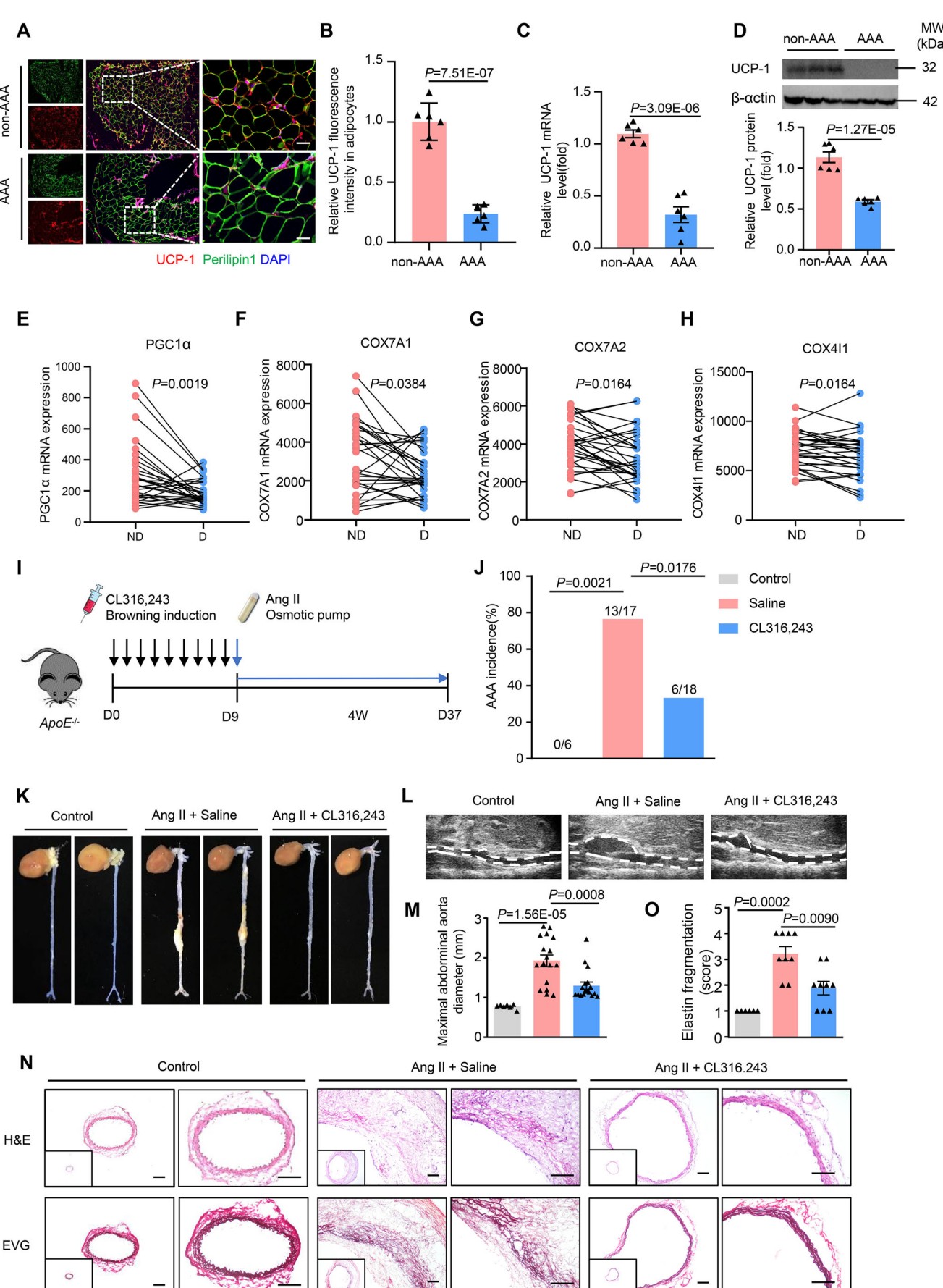

**Figure 1.  Browning of adipose tissue protects against AAA in vivo.**

(A) Representative immunofluorescent staining of uncoupling protein 1 (UCP-1, red) and Perilipin 1 (green) in perivascular adipose tissue (PVAT) in abdominal aortic aneurysm (AAA) patients or healthy control ($n = 6$). Scale bar = 50 μm. (B) Quantification of UCP-1 staining in (A). (C) Q-PCR analysis of UCP-1 mRNA level in PVAT of AAA patients and control ($n = 6$). (D) Western blot analysis of UCP-1 protein level in PVAT of AAA patients and control ($n = 6$). (E–H) Expression of browning-related genes in perivascular adipose tissues of dilated vs non-dilated AAA patients (GSE119717, $n = 30$). ND non-dilated, D dilated. (I) Experimental design. Twenty-week-old male ApoE$^{-/-}$ mice were injected with saline or CL316,243 and then infused with Ang II (1000 ng/kg/min) for 28 days. (J) AAA incidence. (K) Representative photographs of the abdominal aorta taken during sacrifice. (L) Representative ultrasound images of abdominal aortas 4 weeks after Ang II infusion. (M) Quantification of maximal abdominal aortic diameter ($n = 6, 17, 18$). (N) Hematoxylin and eosin (H&E) and Elastica Van Gieson (EVG) staining of abdominal aortic sections. Scale bar = 50 μm. (O) Quantification of elastin fragmentation ($n = 6, 9, 9$). Paired or unpaired Student's $t$ test was used for comparing differences between two groups. Fisher's exact test was used to analyze differences in AAA incidence. One-way ANOVA and Kruskal–Wallis H test were used for multiple group comparisons. Source data are available online for this figure.

with flow cytometry analysis indicated that direct CL316,243 treatment did not change H$_2$O$_2$-induced VSMC apoptosis (Fig. EV5A,B), suggesting that CL316,243 has an indirect effect on VSMCs. We further cultured human VSMCs with conditioned medium obtained from adipocytes that have been treated with PBS or CL316,243, and then examined VSMC apoptosis (Fig. 3A). As shown in Fig. EV6A,B, CL316,243 treatment robustly induced adipocyte browning, as indicated by dramatical increase of Ucp-1 expression. Notably, conditioned medium derived from CL316,243-treated adipocytes significantly inhibited VSMC apoptosis induced by either H$_2$O$_2$ or TNFα plus CHX, as measured by Annexin-V/PI assay (Fig. 3B–E). To further investigate the apoptotic process, we detected the level of selected apoptosis markers- the cleavage of poly ADP-ribose polymerase (PARP) and Caspase 3. The results showed that conditioned medium derived from CL316,243-treated adipocytes significantly decreased the cleavage of both PARP and Caspase 3 (Fig. 3F–H). Collectively, these results indicated that the secretome of browning adipocytes inhibits VSMC apoptosis.

## Identification of FSTL1 as a novel batokine inhibiting VSMC apoptosis

To explore the potential protective factors secreted by browning adipocytes, conditioned medium from PBS or CL316,243-treated adipocytes was subjected to liquid chromatography with tandem mass spectrometry (LC-MS/MS) analysis. A total of 106 proteins were identified to be differentially expressed in the medium of CL316,243-treated adipocytes, with a criterion of 1.5-fold change and $P < 0.05$, of which 64 were upregulated and 42 were downregulated (Fig. 4A). After subcellular localization classification of these proteins using UniProt database, 28 proteins classified as "secreted" were selected, with 18 upregulated and 10 downregulated after CL316,243 treatment (Fig. 4B,C). The upregulated secreted proteins were then subjected to cluster analysis (Fig. 4D). Among the 18 proteins that were upregulated in browning adipocytes, Follistatin-like 1 (FSTL1) emerged as a protein of particular interest because of its anti-apoptotic function in multiple cell types (Liang et al, 2014; Ogura et al, 2012; Ouchi et al, 2008; Wei et al, 2015). By western blot analysis, we confirmed that FSTL1 was dramatically increased in the conditioned medium of CL316,243-treated adipocytes (Fig. 4E). We further analyzed FSTL1 expression in adipose tissues in vivo. As shown in Fig. 4F, administration of browning agent CL316,243 significantly increased FSTL1 expression in both epididymal white adipose tissues (eWAT) and subcutaneous white adipose tissues (sWAT). In addition, we found that FSTL1 levels were significantly decreased in the plasma of

AAA patients (Fig. 4G). After adjusting for age, sex, smoking status, drinking status, body mass index, systolic and diastolic blood pressure, heart rate, LDL, HDL, hypertension, and diabetes, plasma FSTL1 levels remained significantly associated with AAA after adjustment (OR 0.94, 95% CI 0.90–0.96, $P < 0.001$). Together, all these data suggested that FSTL1 may be a novel vessel-protective batokine.

Next, we investigated whether FSTL1 inhibits VSMC apoptosis. The results showed that recombinant FSTL1 treatment significantly inhibits either TNFα plus CHX or H$_2$O$_2$-induced VSMC apoptosis, as measured by Annexin-V/PI staining and flow cytometry analysis (Fig. 4H,I). Consistently, recombinant FSTL1 treatment significantly decreased the cleavage of PARP and Caspase 3 (Fig. 4J,K). Together, our results suggested that the secreted FSTL1 inhibits VSMC apoptosis.

## FSTL1 inhibits VSMC apoptosis via receptor DIP2A-mediated AKT activation

DIP2A is a cell surface receptor for FSTL1, which has been reported to mediate the protective effects of FSTL1 in endothelial cells (Ouchi et al, 2010). To investigate whether DIP2A mediates the protective effects of FSTL1 in VSMCs, we silenced DIP2A in VSMCs and then treated the cells with recombinant FSTL1. The results showed that DIP2A deficiency abrogated the inhibitory effect of FSTL1 on VSMC apoptosis induced by either TNFα plus CHX or H$_2$O$_2$ (Fig. 5A,B). In addition, decreased cleavage of PARP and Caspase 3 mediated by FSTL1 was also reversed in DIP2A-deficient VSMCs (Fig. 5C,D).

To elucidate the mechanism underlying the effects of FSTL1 on VSMCs, we conducted RNA sequencing in H$_2$O$_2$-induced VSMCs treated with or without FSTL1. Kyoto Encyclopedia of Genes and Genomes (KEGG) analysis showed that the upregulated genes in FSTL1-treated VSMCs were enriched in the PI3K-AKT signaling pathway (Fig. 5E). We further confirmed the activation of AKT signaling by western blot analysis. As shown in Fig. 5F, FSTL1 treatment significantly enhanced H$_2$O$_2$-induced AKT phosphorylation, whereas it had no effect on ERK phosphorylation. In addition, FSTL1-mediated AKT phosphorylation was abolished in DIP2A-deficient VSMCs (Fig. 5G). Moreover, pharmacologic inhibition of AKT signaling by MK-2206, an allosteric AKT inhibitor, abrogated the inhibitory effect of FSTL1 on VSMC apoptosis (Fig. 5H). In addition, decreased cleavage of PARP and Caspase 3 mediated by FSTL1 was also abrogated by AKT inhibition (Fig. 5I). Together, these data suggested that the protective effect of FSTL1 in VSMCs is through the DIP2A/AKT pathway.

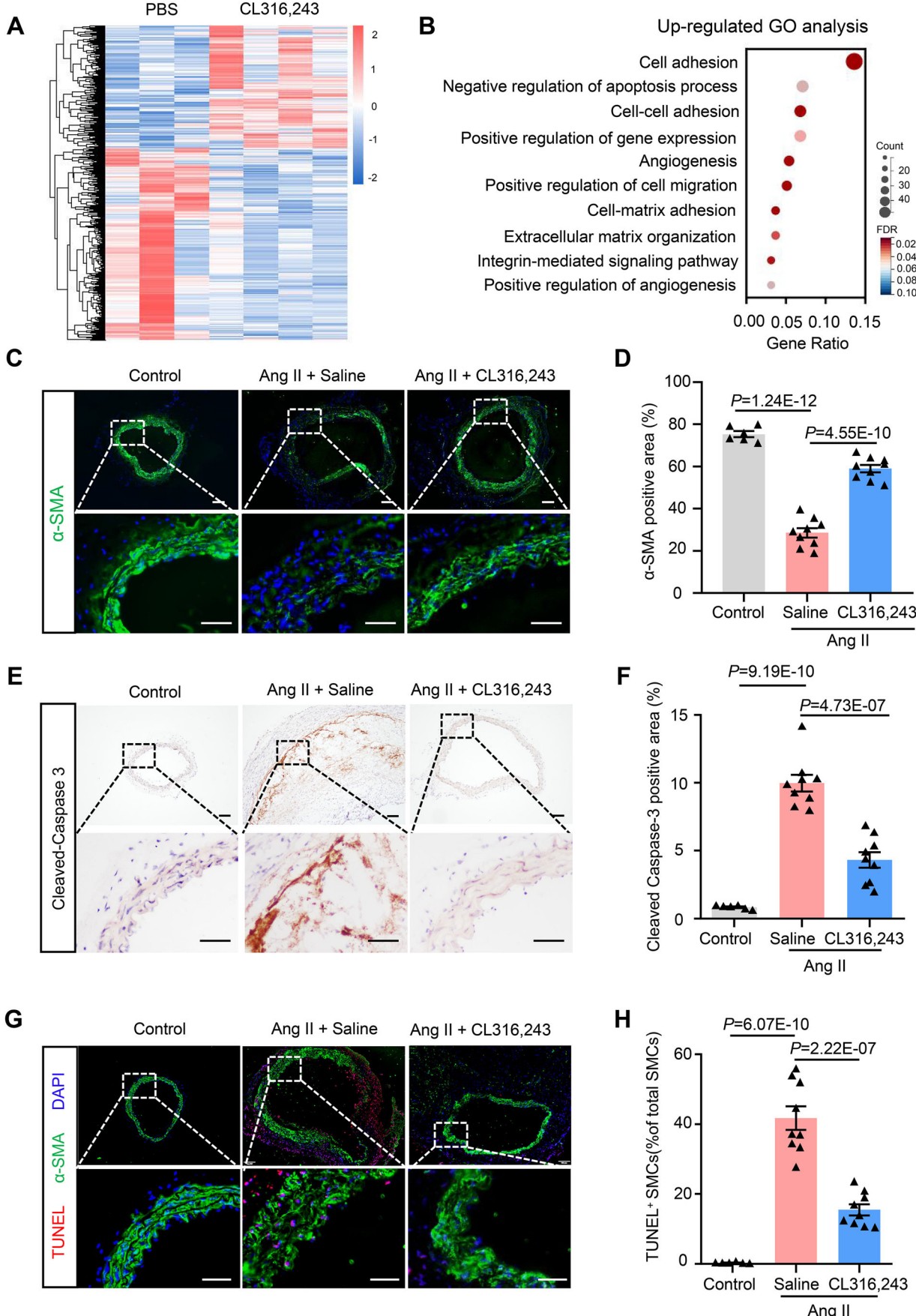

◄ **Figure 2. Browning of adipose tissue inhibits VSMC apoptosis in AAA.**

(A) Heatmap of differentially expressed genes in aortic tissue. (B) Gene ontology (GO) analysis. (C) Representative immunofluorescent staining of alpha smooth muscle actin (α-SMA) in suprarenal abdominal aortic sections (scale bar, 50 μm). (D) Quantification of α-SMA positive area as in (C) ($n = 6, 9, 9$). (E) Representative immunohistochemical staining of Cleaved-Caspase 3 in suprarenal abdominal aortic sections (scale bar, 50 μm). (F) Quantification of Cleaved-caspase 3-positive area as in (E) ($n = 6, 9, 9$). (G) Representative terminal deoxynucleotidyl transferase dUTP nick end labeling (TUNEL) and α-SMA staining of suprarenal aortic sections (scale bar, 50 μm). (H) Quantification of TUNEL-positive SMCs ($n = 6, 9, 9$). One-way ANOVA was used to determine statistical difference. Source data are available online for this figure.

## Adipocyte-specific FSTL1 deficiency abrogates the protective effect of browning induction against AAA

To determine whether the protective effect of browning induction against AAA is through FSTL1 secreted by adipocytes, we constructed adipocyte-specific FSTL1-deficient mice. Briefly, FSTL1$^{flox/flox}$ mice were injected with AAV-Adipoq-Cre or AAV-control and then administered with CL316,243 to induce adipose tissue browning, followed by CaPO4 application to induce AAA formation (Fig. 6A). We first confirmed that FSTL1 expression after CL316,243 administration was significantly repressed by AAV-Adipoq-Cre injection in adipose tissues such as eWAT, sWAT, and PVAT (Fig. EV7A), but not in other tissues, such as heart, liver, and kidney (Fig. EV7B). Notably, we found that browning induction by CL316,243 significantly reduced aortic dilation and decreased the maximal diameter of the abdominal aorta in the AAV-Control group, whereas it showed minimal effects in adipocyte-specific FSTL1-deficient mice (Fig. 6B–D). In addition, browning induction significantly decreased arterial wall thickness and elastin fragmentation in AAV-control mice, whereas adipocyte FSTL1 deficiency abrogated these effects (Fig. 6E,F). Moreover, the increased SMC content by browning induction was also abrogated (Fig. 6G,H). Together, these results suggested that the protective effect of browning induction is through adipocyte-derived FSTL1.

## Hydrogel-patched recombinant FSTL alleviates AAA development

To further determine whether local FSTL1 treatment impacts AAA development, Pluronic F-127 Hydrogel containing recombinant FSTL1 was applied around the abdominal aorta after CaPO4 treatment (Fig. 7A). The results showed that FSTL1-containing hydrogel treatment significantly decreased CaPO4-induced aortic dilation and the maximal diameter of the abdominal aorta (Fig. 7B–D). Histological analysis further indicated that FSTL1-containing hydrogel treatment significantly decreased elastin fragmentation (Fig. 7E,F). In addition, TUNEL staining revealed that apoptosis of SMC in the abdominal aorta was significantly decreased by FSTL1-containing hydrogel treatment (Fig. 7G,H). All these data suggested that local treatment of recombinant FSTL1 around the abdominal aorta could alleviate AAA development.

## Systemic FSTL1 infusion attenuates AAA development

We next explored the therapeutic potential of FSTL1 infusion in attenuating AAA development. Starting from the 7th day after Ang II modeling, ApoE−/− mice were intraperitoneally injected with recombinant FSTL1 protein at a dose of 100 μg/kg or saline control every other day (Fig. 8A). After 3 weeks of treatment, the results showed that FSTL1 infusion significantly decreased the maximal diameter of the abdominal aorta (Fig. 8B,C). In addition, FSTL1 infusion significantly decreased Ang II-induced elastin fragmentation and increased vessel integrity (Fig. 8D,E). Moreover, FSTL1 infusion significantly increased SMC content as indicated by more SMA staining (Fig. 8F,G). The apoptosis of SMC was also significantly inhibited, as indicated by decreased cleaved-caspase 3 staining and co-staining of TUNEL and SMA (Fig. 8H–K). These results suggested that systemic FSTL1 infusion effectively inhibited VSMC apoptosis and restricted AAA development.

## Discussion

Abdominal aortic aneurysm (AAA) is a life-threatening aortic disease without pharmacological approaches. Adipose tissue is composed of white adipose tissue (WAT) and brown adipose tissue (BAT), which differ morphologically and functionally. Multiple lines of evidence demonstrate that there is a significant expansion of white adipose tissue with abdominal aortic aneurysm development (Cronin et al, 2013; Stackelberg et al, 2013), whereas the prevalence and amount of brown adipose tissue declines with advancing age (Cypess et al, 2009), which is known to increase the risk of AAA(Quintana et al, 2019b). Although adipose tissue has been implicated in AAA development, little attention has been paid to its remodeling in AAA progression. Here, we found that AAA patients have a reduced browning level in perivascular adipose tissue, and browning induction in white adipose tissue significantly attenuated AAA development in mice. Our study suggested that remodeling the white adipose tissue by "re-browning" may be beneficial to prevent or slow down AAA progression.

Interestingly, while we observed significantly decreased browning levels in AAA patients, the expression of browning-associated genes was increased in Ang II-induced mouse models (Fig. EV8). Since Ang II is known to induce inflammatory responses that require increased energy expenditure, this elevated browning may represent a compensatory mechanism against Ang II-induced inflammation and metabolic stress. However, as AAA progresses, chronic inflammation and metabolic dysfunction may eventually overwhelm this protective response, leading to a decline in browning levels.

In the present study, β3 adrenergic receptor agonist treatment was used to induce white adipose tissue browning. Indeed, there are some other classical ways for browning induction, such as cold exposure and exercise, during which norepinephrine will be

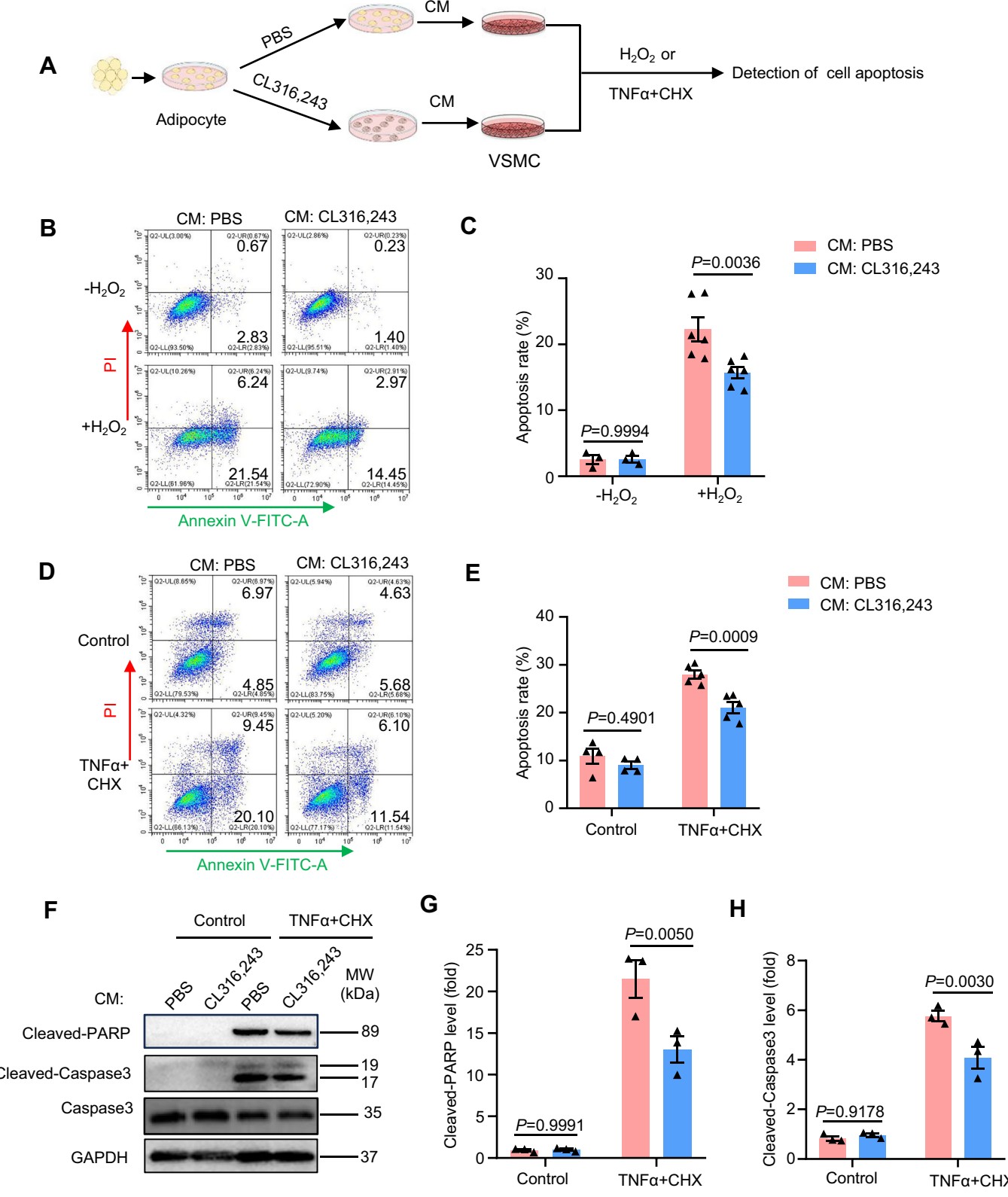

released by the activated sympathetic nervous system and binds to β3 adrenergic receptor on adipocytes to induce browning (Cannon et al, 2004; Seals et al, 1991; van Marken Lichtenbelt et al, 2009). However, it is worth noting that increased sympathetic excitability can also cause accelerated heartbeat and high blood pressure, which

are important risk factors for AAA. Therefore, we should be cautious about cold exposure and exercise training as browning induction methods for practical use against AAA. Recently, some special dietary patterns such as intermittent fasting and caloric restriction have also been demonstrated to induce adipose tissue

**Figure 3. Secretome of browning adipocytes inhibits VSMC apoptosis.**

(A) Experimental design. Primary adipocytes were treated with PBS or CL316,243 for 24 h, and the culture medium (CM) was then collected and incubated with vascular smooth muscle cells (VSMCs), which further underwent $H_2O_2$ or TNFα plus cycloheximide (CHX) treatment to induce apoptosis. (B) Flow cytometric analysis of Annexin V-FITC staining in VSMCs treated with adipocyte CM and $H_2O_2$. (C) Quantification of apoptosis as in B (without $H_2O_2$, $n = 3$; with $H_2O_2$, $n = 6$). (D) Flow cytometric analysis of Annexin V-FITC staining in VSMCs treated with adipocyte CM and TNFα plus CHX. (E) Quantification of apoptosis as in (D) (without TNFα + CHX, $n = 4$; with TNFα + CHX, $n = 5$). (F) Western blot analysis of cleaved-PARP and cleaved-Caspase 3 in VSMCs treated with adipocyte CM and TNFα plus CHX. (G) Quantification of cleaved-PARP level ($n = 3$). (H) Quantification of cleaved-Caspase 3 level ($n = 3$). Two-way ANOVA was used to determine statistical difference. Source data are available online for this figure.

browning (Fabbiano et al, 2016; Lin et al, 2021). Interestingly, a previous study demonstrated that calorie restriction protects against experimental AAA in mice (Liu et al, 2016). Therefore, it is possible that calorie restriction may also exert its protection via adipose tissue browning.

Browning adipocytes are known to regulate energy metabolism and have thus been implicated for therapeutic exploitation for metabolic diseases, including Type 2 diabetes mellitus (T2DM) and obesity (Liu et al, 2013; Stanford et al, 2013). However, in our current study, although we observed a decrease in AAA incidence and development by browning induction, there was no significant change in metabolic parameters such as body weight and blood glucose levels, suggesting that adipose tissue browning may protect against AAA in a metabolism-independent manner. Interestingly, we found that adipocytes after brown induction dramatically increase the secretion of Follistatin-like 1 (FSTL1), which directly inhibits VSMC apoptosis and AAA development. Importantly, adipocyte-specific FSTL1 deficiency abrogates the protective effect of browning induction against AAA These studies suggested that the secretory function of browning adipocytes may play an essential role in protecting vascular cells and suppressing AAA development. However, a limitation of the current in vitro study is the use of human VSMCs to investigate the effects of mouse adipocyte-derived secretory factors. Future studies should validate these findings in a syngeneic system (e.g., mouse adipocytes co-cultured with primary mouse aortic VSMCs).

FSTL1 is a secreted glycoprotein and has previously been implicated to be a potential prognostic marker for cardiovascular diseases (Widera et al, 2012; Widera et al, 2009). Mark Gorelik et al found that the plasma levels of FSTL1 were elevated in toddler patients with acute Kawasaki disease (KD) (Gorelik et al, 2012), suggesting its level may be associated with acute arterial injury and coronary artery aneurysms (CAA) formation. However, considering the relatively small number of patients in the CAA group (only 7 cases) and the difference between CAA and AAA, the relationship between FSTL1 level and AAA development remains unclear. In this study, we found that plasma levels of FSTL1 were significantly lower in AAA patients, suggesting that reduced levels of FSTL1 may be associated with AAA development. However, whether the FSTL1 level could predict AAA development still needs further investigation.

Many recent studies indicated that FSTL1 not only serves as a molecular marker but also possesses important functions in the development of cardiovascular diseases. FSTL1 insufficiency could result in deformed mitral valves (Prakash et al, 2017), atrial and venous wall fibrosis (Jiang et al, 2020), and pulmonary vasculature defects (Tania et al, 2017). In addition, FSTL1 deletion exacerbates

hypoxia-induced pulmonary hypertension(Zhang et al, 2017), wire-induced arterial injury (Miyabe et al, 2014), pressure overload-induced hypertrophy and cardiac failure (Shimano et al, 2011), etc. Importantly, FSTL1 overexpression or reconstitution has been reported to prevent myocardial infarction and cardiomyopathy (Lu et al, 2024; Oshima et al, 2008; Wei et al, 2015), reduce neointimal formation (Miyabe et al, 2014), and prevent against aortic dissection (Li et al, 2025). Many of the functions of FSTL1 in cardiovascular diseases are associated with its roles in cardiomyocytes and vascular cells. However, accumulating evidence also suggests a role of FSTL1 in adipocytes, as its deficiency could impair brown adipose tissue development and brown adipose tissue thermogenesis(Fang et al, 2019; Guggeri et al, 2024). Hereby, proteomic analysis of the culture medium from browning adipocytes, we identified FSTL1 as a vessel-protective adipokine. Importantly, supplementation of recombinant FSTL1 either at the beginning or during AAA progression significantly attenuates AAA development, suggesting its great potential for intervening AAA.

Notably, our proteomic analysis revealed some other secretory proteins alongside FSTL1 in the browning adipocyte medium. Periostin is an extracellular matrix protein known to regulate VSMC migration and osteoblastic switch, and has been suggested to promote atherosclerosis and vascular calcification (Alesutan et al, 2022; Li et al, 2006; Sun et al, 2021). Thrombospondin-4 is a member of the extracellular calcium-binding protein family and is linked to cell adhesion and migration (Lv et al, 2016), and has been reported to promote VSMC proliferation, local inflammation, atherogenesis, and restenosis(Frolova et al, 2010; Lv et al, 2016; Stenina et al, 2003). Given our focus on identifying protective mediators of adipocyte-VSMC crosstalk, we prioritized targets with clearer therapeutic potential and thus excluded periostin and thrombospondin-4. It is worth noting that serpin superfamily proteins, Serpin A9 and SerpinA3N, were also detected. Given their anti-apoptotic functions in neurological contexts and tumor milieus (Kummer et al, 2007; Zhang et al, 2022), investigation of their potential protective roles in AAA would be of interest in the future study. Separately, transcriptomic analysis of post-browning aortic tissues also revealed downregulation of genes associated with synaptic transmission (Fig. EV9), suggesting a possible link between browning and neurogenic pathways.

In summary, our study demonstrates a protective role of adipose tissue browning in AAA development. Mechanistically, we found that browning adipocytes secreted FSTL1, which inhibited VSMC apoptosis through DIP2A/AKT signaling. Our findings indicate the therapeutic potential of adipose tissue browning induction and FSTL1 supplementation for treating AAA.

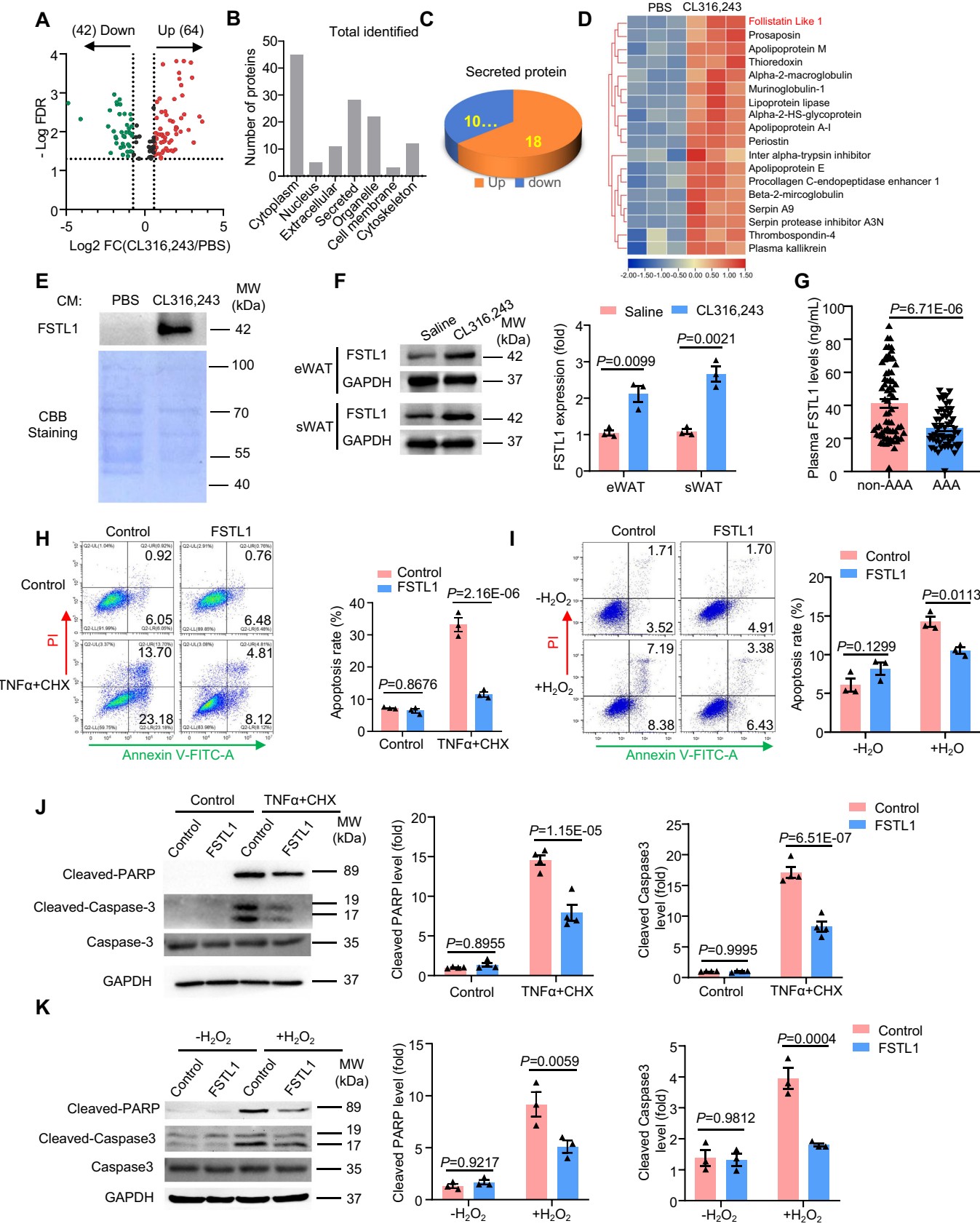

◀ **Figure 4. Identification of FSTL1 as a novel batokine inhibiting VSMC apoptosis.**

(A) Volcano plot of the proteins identified by liquid chromatography with tandem mass spectrometry (LC-MS/MS) ($n = 3$). (B) Subcellular localization categorization of differentially expressed proteins based on subcellular localization. (C) Pie-chart of the number of upregulated and downregulated secreted proteins. (D) Heatmap of the upregulating secreted proteins. (E) Western blot analysis of FSTL1 in CM of adipocytes treated with PBS or CL316,243 for 24 h. Coomassie brilliant blue (CBB) staining was used as a loading control. (F) Western blot analysis of FSTL1 expression in epididymal white adipose tissue (eWAT) and subcutaneous white adipose tissue (sWAT) ($n = 3$). (G) ELISA analysis of FSTL1 in the plasma of control ($n = 68$) and AAA patients ($n = 55$). Student's $t$ test was used to determine statistical difference. (H, I) Flow cytometric analysis of Annexin V-FITC staining in VSMCs treated with recombinant FSTL1 (100 ng/mL) and stimulated with TNFα plus CHX (H) or $H_2O_2$ (I) ($n = 3$). (J, K) Western blot analysis of cleaved-PARP and cleaved-Caspase 3 in VSMCs treated with recombinant FSTL1 and stimulated with TNFα plus CHX (J) ($n = 4$) or $H_2O_2$ (K) ($n = 3$). Two-way ANOVA was used to determine statistical difference. Source data are available online for this figure.

# Methods

### Reagents and tools table

| Reagent/resource | Reference or source | Identifier or catalog number |
| --- | --- | --- |
| **Experimental models** | | |
| ApoE$^{-/-}$ mice | Gempharmatech Co., Ltd | Cat# T001458-11 |
| Fstl1$^{flox/flox}$ mice | Gempharmatech Co., Ltd | Cat#T018356 |
| C57BL/6 J mice | Gempharmatech Co., Ltd | Cat# C57 N000013 |
| HASMC | Sunn Cell Biotech | Cat#SNL-683 |
| **Antibodies** | | |
| Caspase-3 Antibody (Diluted at 1:1000 for WB) | Cell Signaling Technology | Cat#9662 |
| Cleaved Caspase-3 (Asp175) Antibody(Diluted at 1:1000 for WB; 1:200 for IHC) | Cell Signaling Technology | Cat#9661 |
| Cleaved PARP (Asp214) (D6X6X) Rabbit mAb (Diluted at 1:1000 for WB) | Cell Signaling Technology | Cat#94885 |
| alpha Smooth Muscle Actin antibody (Diluted at 1:200 for IHC and IF) | GeneTex | Cat#GTX100034 |
| FSTL1 Polyclonal antibody (Diluted at 1:2000 for WB) | Proteintech | Cat#20182-1-AP |
| Perilipin 1 Polyclonal antibody (Diluted at 1:200 for IF) | Proteintech | Cat# 27716-1-AP |
| KLF4 Polyclonal antibody (Diluted at 1:200 for IF) | Proteintech | Cat#11880-1-AP |
| Osteopontin Recombinant antibody (Diluted at 1:200 for IF) | Proteintech | Cat# 83341-1-RR |
| DIP2A Antibody - BSA Free (1.0 μg/mL for WB) | Novus | Cat#NBP1-56909 |
| GAPDH (D16H11) XP® Rabbit mAb (Diluted at 1:2000 for WB) | Cell Signaling Technology | Cat #5174 |
| Anti-UCP1 Rabbit polyclonal antibody (Diluted at 1:1000 for WB) | Abcam | Cat#ab10983 |
| Anti-UCP1 Rabbit antibody (Diluted at 1:400 for IF) | Sigma-Aldrich | Cat# U6382 |
| Phospho-Akt (Ser473) Antibody (Diluted at 1:1000 for WB) | Cell Signaling Technology | Cat #9271 |
| Akt Antibody (Diluted at 1:1000 for WB) | Cell Signaling Technology | Cat#9272 |

| Reagent/resource | Reference or source | Identifier or catalog number |
| --- | --- | --- |
| Phospho-p44/42 MAPK (Erk1/2) (Thr202/Tyr204) (D13.14.4E) XP® Rabbit mAb(Diluted at 1:1000 for WB) | Cell Signaling Technology | Cat#4370 |
| Mouse Actin Monoclonal antibody (Diluted at 1:2000 for WB) | Santa Cruz Biotechnology | Cat# sc-58673 |
| Alpha Tubulin Polyclonal antibody (Diluted at 1:5000 for WB) | Proteintech | Cat#11224-1-AP |
| HRP-Affinipure Goat Anti-Mouse IgG(H + L) (Diluted at 1:2000 for WB; 1:200 for IHC) | Proteintech | Cat#SA00001-1 |
| HRP-conjugated Goat Anti-Rabbit IgG(H + L) (Diluted at 1:2000 for WB; 1:200 for IHC) | Proteintech | Cat#SA00001-2 |
| Goat anti-Rat IgG (H + L) Cross-Adsorbed Secondary Antibody, Alexa Fluor™ 555 (Diluted at 1:400 for IF) | Invitrogen | Cat#A21434 |
| Donkey anti-Rabbit IgG (H + L) Highly Cross-Adsorbed Secondary Antibody, Alexa Fluor™ 488 (Diluted at 1:400 for IF) | Invitrogen | Cat#A21206 |
| Goat anti-Rabbit IgG (H + L) Cross-Adsorbed Secondary Antibody, Alexa Fluor™ 568 (Diluted at 1:400 for IF) | Invitrogen | Cat#A-11011 |
| **Oligonucleotides and other sequence-based reagents** | | |
| PCR Primers | This study | "Methods" |
| siRNA oligos | This study | "Methods" |
| **Chemicals, enzymes, and other reagents** | | |
| PhosSTOP™ | Roche | Cat#4906845001 |
| cOmplete™ Mini EDTA-free | Roche | Cat# 4693159001 |
| One-Step TUNEL Apoptosis Detection Kit | Beyotime | Cat#C1090 |
| Weigert elastic fiber staining solution | leagene | Cat#DC0064 |
| Hematoxylin | Servicebio | Cat#G1004-500ML |
| Eosin | Servicebio | Cat#G1001-500ML |
| Bovine serum albumin | Biofroxx | Cat#4240GR100 |
| DAB Horseradish Peroxidase Color Development Kit | ZSGB-BIO | Cat#ZLI-9017 |
| Skim milk | Biofroxx | Cat# 1172GR500 |

| Reagent/resource | Reference or source | Identifier or catalog number |
|---|---|---|
| Human Follistatin Like Protein 1 (FSTL1) ELISA Kit | bioswamp | Cat#HM11377 |
| Osmotic minipumps | Alzet | Cat#2004 |
| Angiotensin II (human), Vasoconstrictor peptide | Abcam | Cat#ab120183 |
| VECTASHIELD® Antifade Mounting Medium with DAPI/ VECTASHIELD | Vector | Cat#H-1200-10 |
| Collagenase I | Biofroxx | Cat# 1904 |
| Dulbecco's Modified Eagle Medium/Nutrient Mixture F-12 | EBYBIO | Cat#C11330500BT |
| Insulin (From Bovine) | Yeasen | Cat#40107ES25 |
| 3,3',5-Triiodo-Lthyronine | Sigma-Aldrich | Cat#T2877 |
| Fetal bovine serum | GIBCO | Cat#10099141 C |
| Penicillin/streptomycin | GIBCO | Cat# 15140-122 |
| Indomethacin | Sigma-Aldrich | Cat# I7378 |
| Dexamethasone | Sigma-Aldrich | Cat#D4902 |
| 3-Isobutyl-1-methylxanthine | Sigma-Aldrich | Cat# I5879 |
| Rosiglitazone | Sigma-Aldrich | Cat# R2408 |
| CL316,243 | Sigma-Aldrich | Cat# C5976 |
| Phosphate Buffered Saline | HyClone | Cat# SH30256.01B |
| DMEM basic | ThermoFisher | Cat# C11995500BT-1 |
| TNFα | ProSpec-Tany | Cat#CYT-223 |
| Cycloheximide | GLPBIO | Cat# GC17198 |
| Hydrogen peroxide solution 3% | Sigma-Aldrich | Cat#88597 |
| Lipofectamine RNAiMAX | Invitrogen™ | Cat# 13778150 |
| Trizol reagent | Invitrogen | Cat# 15596018CN |
| Hifair® II 1st Strand cDNA Synthesis Kit | Yeasen | Cat#11121ES60 |
| ChamQ Universal SYBR qPCR Master Mix | Vazyme | Cat# Q711-02 |
| Trichloroacetic Acid (TCA) Precipitation Kit | KeyGEN Biotech | Cat#KGP5100 |
| Annexin V-FITC/PI staining Kit | KeyGEN Biotech | Cat#KGA1102-100 |
| NEB Next Ultra RNA Library Prep Kit for Illumina | NEB | Cat#E7760S |
| UltraBio™ Oligo (dT)25 Magnetic Beads | Aladdin | Cat#O751564 |
| Agilent 2100 RNA nano 6000 assay kit | Agilent | Cat# 5067-1511 |
| Recombinant Human FSTL1 | PeproTech | Cat#120-51-50 |
| PierceTM Quantitative Colorimetric Peptide Assay | Invitrogen | Cat# 23275 |
| PageRuler™ Unstained Protein Ladder | Thermo Fisher | Cat#26616 |
| BCA Protein Assay Kit | ThermoFisher | Cat# 23235 |
| 0.25% Trypsin-EDTA | ThermoFisher | Cat#25200-072 |
| Pluronic® F-127 | Sigma-Aldrich | Cat#P2443 |

| Reagent/resource | Reference or source | Identifier or catalog number |
|---|---|---|
| AAV-adipoq-Cre | OBiO Technology (Shanghai) Corp., Ltd. | Cat#M1797 |
| AAV-adipoq-control | OBiO Technology (Shanghai) Corp., Ltd. | Cat#H29035 |
| CaCl$_2$ | GuangZhou Chemical Reagent Factory | Cat# GS0177 |
| Glycine | Sigma-Aldrich | Cat# V900144 |
| Trizma base | Sigma-Aldrich | Cat# V900483 |
| Neutral balsam | Servicebio | Cat#WG10004160 |
| RIPA Lysis Buffer | Beyotime | Cat#P0013B |
| Total cholesterol assay kit | Jiancheng | Cat# A111-1-1 |
| Triglyceride assay kit | Jiancheng | Cat# A110-1-1 |
| ACCU-CHEK Performa test strips | Roche | Cat# 06454011022 |
| Trizma base | Sigma-Aldrich | Cat# V900483 |
| TSA Fluorescence Staining Kit | Servicebio | Cat# G1226-50T |
| Smooth muscle cell medium | FineTest | Cat# C490-WP |
| Proteomics sample pretreatment kit | kefu_tech | Cat#proteomic kit-10 |
| MK2206 | Selleck Chemicals | Cat#S1078 |
| **Software** | | |
| ImageJ Software | https://imagej.net/ij/download.html | |
| FlowJo software Version 10.3 | BD Biosciences | |
| Roche LightCycler® 480 II software | Roche | |
| **Other** | | |
| Leica CM3050S cryostat | Leica | |
| Visitech Systems | BP-2000 | |
| Flow cytometer | BD Biosciences | |
| Agilent 2100 Bioanalyzer | Agilent Technologies | |
| Roche LightCycler 480 II | Roche | |
| NanoPhotometer® | IMPLEN | |
| Illumina Novaseq6000 | Gene Denovo Biotechnology Co. | |
| Upright Fluorescence Microscope | Olympus BX63 | |
| Multimode Reader | BioTek H1 | |
| Upright Microscope | Nikon NI-U | |
| iBright Imaging Systems | Invitrogen iBright CL1000 | |
| Orbitrap Fusion | ThermoFisher | |
| Vevo 2100 echography device | Fujifilm VisualSonics | |
| ACCU-CHEK Performa | Roche | |
| Noninvasive tail-cuff method | Visitech BP-2000 | |

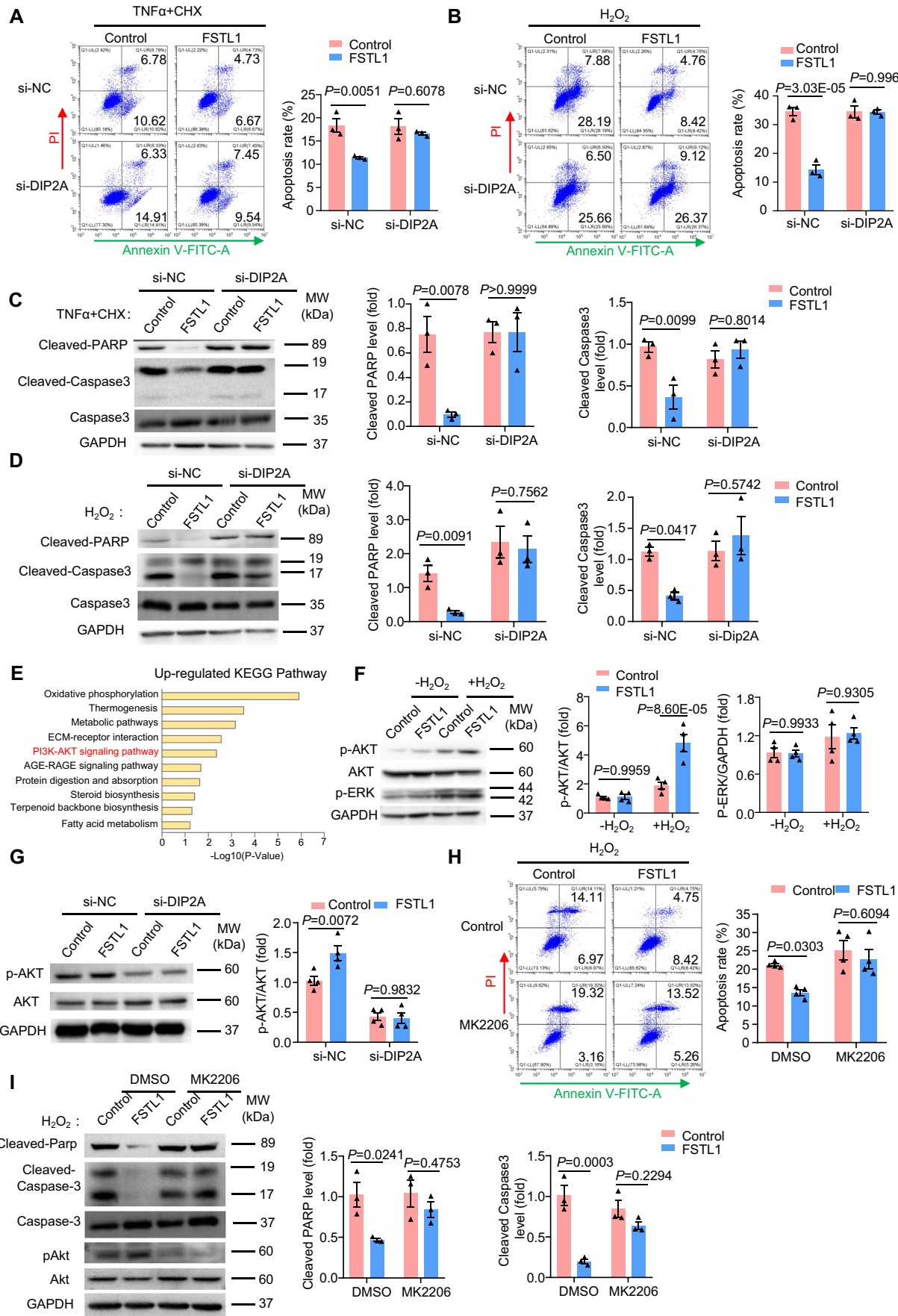

**Figure 5. FSTL1 inhibits VSMC apoptosis via receptor DIP2A-mediated AKT activation.**

(A, B) Flow cytometric analysis of Annexin V-FITC staining in VSMCs silenced with DIP2A and stimulated with TNFα plus CHX (A) or $H_2O_2$ (B) in the absence or presence of FSTL1 ($n = 3$). (C, D) Western blot analysis of cleaved-PARP and cleaved-Caspase 3 in VSMCs silenced with DIP2A and stimulated with TNFα plus CHX (C) or $H_2O_2$ (D) ($n = 3$). (E) Kyoto Encyclopedia of Genes and Genomes (KEGG) analysis of differentially expressed genes in VSMCs treated with/without recombinant FSTL1 and stimulated with $H_2O_2$. (F) Western blot analysis of p-AKT and p-ERK in VSMCs treated with/without recombinant FSTL1 and stimulated with $H_2O_2$ ($n = 4$). (G) Western blot analysis of p-AKT in VSMCs silenced with DIP2A and treated with/without FSTL1 ($n = 4$). (H) Flow cytometric analysis of Annexin V-FITC staining in VSMCs treated with AKT inhibitor MK2206 and stimulated with $H_2O_2$ ($n = 4$). (I) Western blot analysis of cleaved-PARP and cleaved-Caspase 3 in VSMCs treated with AKT inhibitor MK2206 and stimulated with $H_2O_2$ ($n = 3$). Two-way ANOVA was used to determine statistical difference. Source data are available online for this figure.

## Human tissue study

The collection of human tissue and plasma samples was conducted in accordance with the approved protocol by the Sun Yat-sen Memorial Hospital Ethics Committee (SYSKY-2023-1135-01). Informed consent was obtained from all participants, and the experiments conformed to the principles set out in the WMA Declaration of Helsinki and the Department of Health and Human Services Belmont Report. Perivascular adipose tissue of abdominal aortic tissue samples was collected from AAA patients undergoing surgical operations, and control perivascular adipose tissues were obtained from donors undergoing liver transplantation. Plasma samples were collected from patients diagnosed with AAA by abdominal ultrasound. Control participants who were confirmed to be free of AAA by abdominal ultrasound and matched to AAA cases by age (±2 years), sex, and date of hospitalization were selected. Individuals with hereditary syndromes known to cause distinct aortic pathology (e.g., Marfan syndrome, Loeys–Dietz syndrome) were excluded. The samples were collected from participants of Chinese ethnicity, both male and female. The detailed baseline characteristics of all patients are presented in Tables 1 and 2. Publicly available normalized gene expression data were obtained from the Gene Expression Omnibus (GEO) database (accession number: GSE119717).

## Animals

All study protocols for animal experiments were approved by the Institutional Animal Care and Use Committee of Sun Yat-sen Memorial Hospital and conformed to the current National Institutes of Health (NIH) guidelines for Care and Use of Laboratory Animals (AP20220082). ApoE$^{-/-}$ mice, C57BL/6 J mice, and FSTL1$^{flox/flox}$ were purchased from Gempharmatech Co., Ltd. All mice were housed in a controlled environment (20 ± 2 °C, 12-h/12-h light/dark cycle), where they were maintained on a standard chow diet with free access to water. Animals were randomly grouped. All animals used are male due to variability in phenotype. During the feeding period, the body weights of the mice were monitored weekly. In all experiments, mice were anaesthetized by isoflurane inhalation (1.5–2%) followed by euthanasia via cervical dislocation.

## Ang II-induced AAA model

In all, 20-week-old ApoE$^{-/-}$ mice were used for Ang II-induced AAA model. Osmotic minipumps containing Ang II were subcutaneously implanted in the dorsum of the neck at a release rate of 1.44 mg/kg/day for four weeks. Blood pressure was monitored by a noninvasive tail-cuff method. At the end of the 28-day Ang II infusion period, mice underwent vascular ultrasonography imaging and were then euthanized, and the aortas were

dissected, photographed, and measured for maximal abdominal aortic diameter. AAA was defined as ≥50% dilation of the external diameter of the abdominal aorta compared with the normal abdominal aorta in saline-infused mice of the control group, as previously described (Lu et al, 2020).

## CaPO4-induced AAA model

8-week-old C57BL/6 J mice were used for CaPO$_4$-induced AAA model. The infrarenal region of the abdominal aorta was isolated, measured, and surrounded with a small piece of gauze soaked in CaCl$_2$ (0.5 M) for 10 min, replaced with another piece of PBS-soaked gauze for 5 min. Mice in the sham group received a single treatment of PBS-soaked gauze for 15 min. Fourteen days after PBS or CaPO$_4$ treatment, mice were euthanized, the aortic outer diameter was measured, and tissues were collected for analysis.

## Induction of adipose tissue browning

Mice were daily intraperitoneally injected with CL316,243 (1 mg/kg) for 9 days. One day following the final injection, mice were sacrificed for investigation of adipose tissue browning or underwent Ang II or CaPO$_4$ treatment.

## Construction of adipocyte-specific FSTL1-deficient mice

FSTL1$^{flox/flox}$ mice were subcutaneously multipoint injected and intraperitoneally injected with AAV8-adipo-cre or AAV8 control at $5 \times 10^{10}$ vg/mice. Two weeks after AAV injection, FSTL1 expression in different adipose depots and other tissues was investigated.

## FSTL1 treatment

For local FSTL1 treatment, recombinant FSTL1 (0.1 mg/mL) was mixed with Pluronic F-127 hydrogel on ice until fully dispersed. C57BL/6J mice were first treated with CaPO$_4$ at the infrarenal region of the abdominal aorta, and then FSTL1-hydrogel composite was applied to cover the wound after CaPO$_4$ incubation. For systemic FSTL1 treatment, recombinant FSTL1 (100 μg/kg) was intraperitoneally injected into ApoE$^{-/-}$ mice every other day for 3 weeks, starting from day 7 after Ang II induction.

## Histological analysis

The abdominal aorta with maximal diameter were embedded in OCT and sectioned to 6 um thickness using a Leica CM3050S cryostat. Tissue sections were prepared for hematoxylin-eosin (H&E), elastin van Gieson (EVG) staining. The criteria for different grades of elastin fragmentation are mainly based on previous studies (Lu et al, 2020;

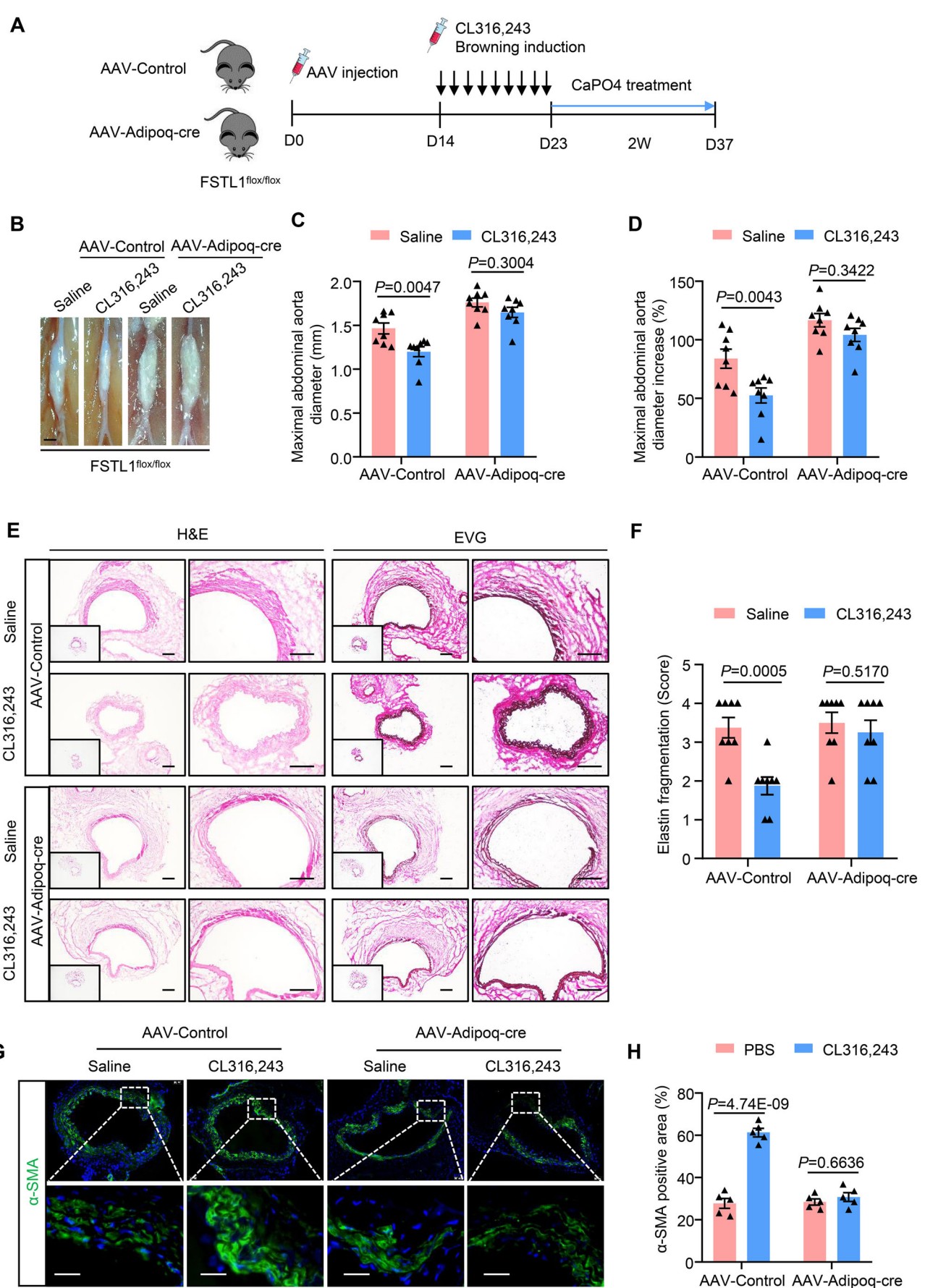

◄ **Figure 6.  Adipocyte-specific FSTL1 deficiency abrogates the protective effect of CL316,243 against AAA.**

(A) Experimental design. AAV-Control or AAV-Adipoq-Cre was first injected into FSTL1$^{fl/fl}$ mice, followed by CL316,243 treatment for 9 days and CaPO4 treatment for another 2 weeks to induce AAA formation. (B) Representative photographs of abdominal aortas (scale bar, 1 mm). (C) Quantification of maximal abdominal aortic diameter ($n = 8$). (D) Quantification of maximal abdominal aortic diameter increase ($n = 8$). (E) Representative H&E and EVG staining images of abdominal aortic sections (scale bar, 50 μm). (F) Quantification of elastin fragmentation ($n = 8$). (G) Representative immunofluorescent staining of α-SMA in suprarenal abdominal aortic sections (scale bar, 25 μm). (H) Quantification of α-SMA positive area as in (C) ($n = 5$). Two-way ANOVA was used to determine statistical differences. Source data are available online for this figure.

Zhang et al, 2024) and are briefly as: 1, no elastin degradation; 2, mild degradation: degradation of less than 25%; 3, moderate degradation: degradation of 25–50%; 4, moderate to severe degradation: degradation of 50–75%; 5, severe degradation: degradation of more than 75%.

For immunohistochemical staining, aortic sections were blocked with 1% bovine serum albumin (BSA) in PBS for 1 h at room temperature. Primary antibodies were then applied onto sections and incubated overnight at 4 °C, followed by HRP-conjugated secondary antibodies before staining with DAB Kit. Hematoxylin was used to counterstain nuclei. Sections incubated with species-matched IgG alone were used as negative controls. Primary antibodies against Cleaved-caspase 3 (CST) and smooth muscle alpha actin were used.

For immunofluorescent staining, aortic sections were blocked with 1% BSA in PBS for 1 h at room temperature. Primary antibodies were then applied onto sections and incubated overnight at 4 °C, followed by Alexa Fluor secondary antibody incubation for 1 h at room temperature. DAPI was used to stain the nuclei. Sections incubated with species-matched IgG alone were used as negative controls. Primary antibody against α-SMA was purchased from GeneTex. For TUNEL staining, aortic sections were stained with the One-Step TUNEL Apoptosis Detection Kit according to the manufacturer's protocol. For all immunofluorescent staining, an anti-fluorescence quencher was applied, and the samples were immediately imaged using a confocal microscope. To allow comparative quantification, fluorescent images of each set of experiments were acquired under the same microscope exposure settings.

Quantification of immunostaining was performed using ImageJ Software. For DAB-stained images, color deconvolution was applied to separate the brown DAB signal from hematoxylin counterstain. For fluorescence images, raw images were first converted to grayscale format. Then the aortic wall area was selected and specific staining signals were defined using thresholding tools. To ensure consistency, the same threshold was applied. Then signal intensity normalized to selected aortic wall area was measured.

## Evaluation of browning

To assess adipose tissue browning, immunohistochemistry or immunofluorescence staining was performed using antibodies against UCP1 (uncoupling protein 1), a hallmark of browning (Wu et al, 2012). Besides, mRNA expression levels of browning-related genes (e.g., PGC1α, COX7A, COX4I) (Fisher et al, 2012; Hattori et al, 2016) were analyzed by quantitative real-time PCR. Protein levels of UCP1 were also assessed by western blot to validate the browning phenotype.

## Primary adipocyte culture and treatment

The inguinal subcutaneous adipose tissue from C57BL/6 J mice was fully cut and digested by collagenase I (1 mg/mL) for 30–40 min at 37 °C, shaking at 150 rpm. Cell suspension was then filtered and centrifuged at 700 × g for 10 min. The pellet was re-suspended in DMEM/F12 complete medium containing 5 μg/mL insulin, 1 nM 3,3′,5-Triiodo-L-thyronine, 10% fetal bovine serum, and 1% Penicillin/Streptomycin. Grow primary adipocytes to 95–97% confluence in complete medium and then induce differentiation by replacing the medium with DMEM/F12 induction medium containing 125 μM Indomethacin, 2 μg/ml Dexamethasone, 0.5 mM 3-Isobutyl-1-methylxanthine, 0.5 μM Rosiglitazone, 10% FBS, and 1% penicillin/streptomycin at 37 °C in 5% $CO_2$. Change the induction medium every 2–3 days until the cells are fully differentiated. The induction medium was freshly made each time. After about 6–7 days of differentiation, cells are fully differentiated to mature fat cells and are filled with oil droplets. The differentiated adipocytes were then treated with CL316,243 or PBS for 24 h, and the conditioned medium was collected for further analysis.

## Smooth muscle cell culture and treatment

Human aortic smooth muscle cells were purchased from Sunncell Biotech and maintained in DMEM, supplemented with 10% FBS and 1% P/S at 37 °C in 5% $CO_2$. Cells have been tested for mycoplasma contamination and were confirmed to be negative. Cells were verified by smooth muscle alpha actin antibody staining and cells between three and eight passages were used. At 80% confluency, cells were treated with $H_2O_2$ for 24 h or with TNF-α and CHX for 6 h to induce apoptosis.

## siRNA transfection

Small interfering RNA (siRNA) against human DIP2A was designed and synthesized by Qingke (Beijing, China). The human DIP2A targeting sequences of siRNA oligos were: sense, 5'-GAU UGCUUGGAAUCACGAA(dT)(dT)-3' and antisense, 5'-UUCGU GAUUCCAAGCAAUC(dT)(dT)-3'. VSMCs were transfected with siRNA (50 nM) via Lipofectamine RNAiMAX according to the manufacturer's protocol.

## RNA isolation and real-time PCR (RT-PCR)

Total RNA was extracted from VSMCs or adipose tissue using Trizol reagent according to the manufacturer's protocols. Subsequently, cDNA was reverse transcribed from 1 μg of total RNA using Hifair II 1st Strand cDNA Synthesis Kit. SYBR Green 2× PCR mix was used according to the manufacturer's instructions. The primer for human UCP-1: forward primer: 5'-AGGTCCAAGGTGAATGCCC-3', reverse primer: 5'-TTACCACAGCGGTGATTGTTC-3'. The primer for human GAPDH: forward primer: 5'-GGAGCGAGATCCCTCC AAAAT-3', reverse primer: 5'-GGCTGTTGTCATACTTCTCAT

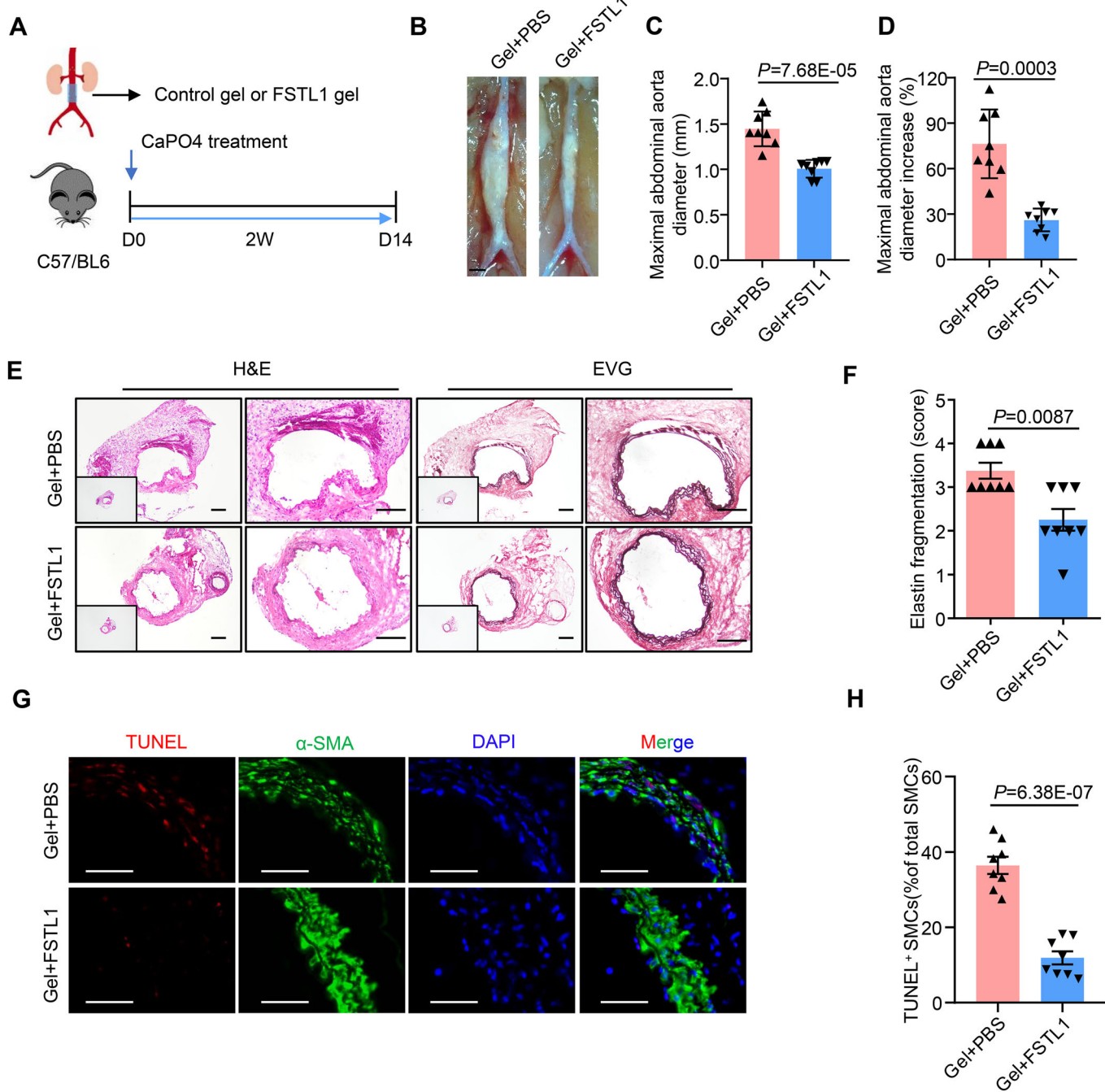

**Figure 7. Hydrogel-patched recombinant FSTL alleviates AAA development.**

(A) Experimental design. Pluronic F-127 hydrogel containing recombinant FSTL1 was applied around the abdominal aorta after CaPO4 treatment in C57/BL6 mice. (B) Representative photographs of the abdominal aorta (scale bar, 1 mm). (C) Quantification of maximal abdominal aortic diameters ($n = 8$). (D) Quantification of maximal abdominal aortic diameter increase ($n = 8$). (E) Representative H&E and EVG staining images of abdominal aortic sections (scale bar, 50 μm). (F) Quantification of elastin fragmentation ($n = 8$). (G) Representative TUNEL and α-SMA staining of abdominal aortic sections (scale bar, 25 μm). (H) Quantification of TUNEL-positive SMCs ($n = 8$). Student's $t$ test and Mann–Whitney $U$ tests were used to determine statistical difference. Source data are available online for this figure.

GG-3'. The primer for mouse UCP-1: forward primer: 5'-GGATTGGCCTCTACGACTCA-3', reverse primer: 5'-TAAGCCGG CTGAGATCTTGT-3'. The primer for mouse GAPDH: forward primer: 5'-TGACCTCAACTACATGGTCTACA-3', reverse primer: 5'-CTTCCCATTCTCGGCCTTG-3'.

## Secreted protein extraction

Secreted protein from the culture medium of adipocytes was extracted using the Trichloroacetic Acid precipitation Kit. Briefly, 200 μL of the culture medium supernatant was added to 50 μL of

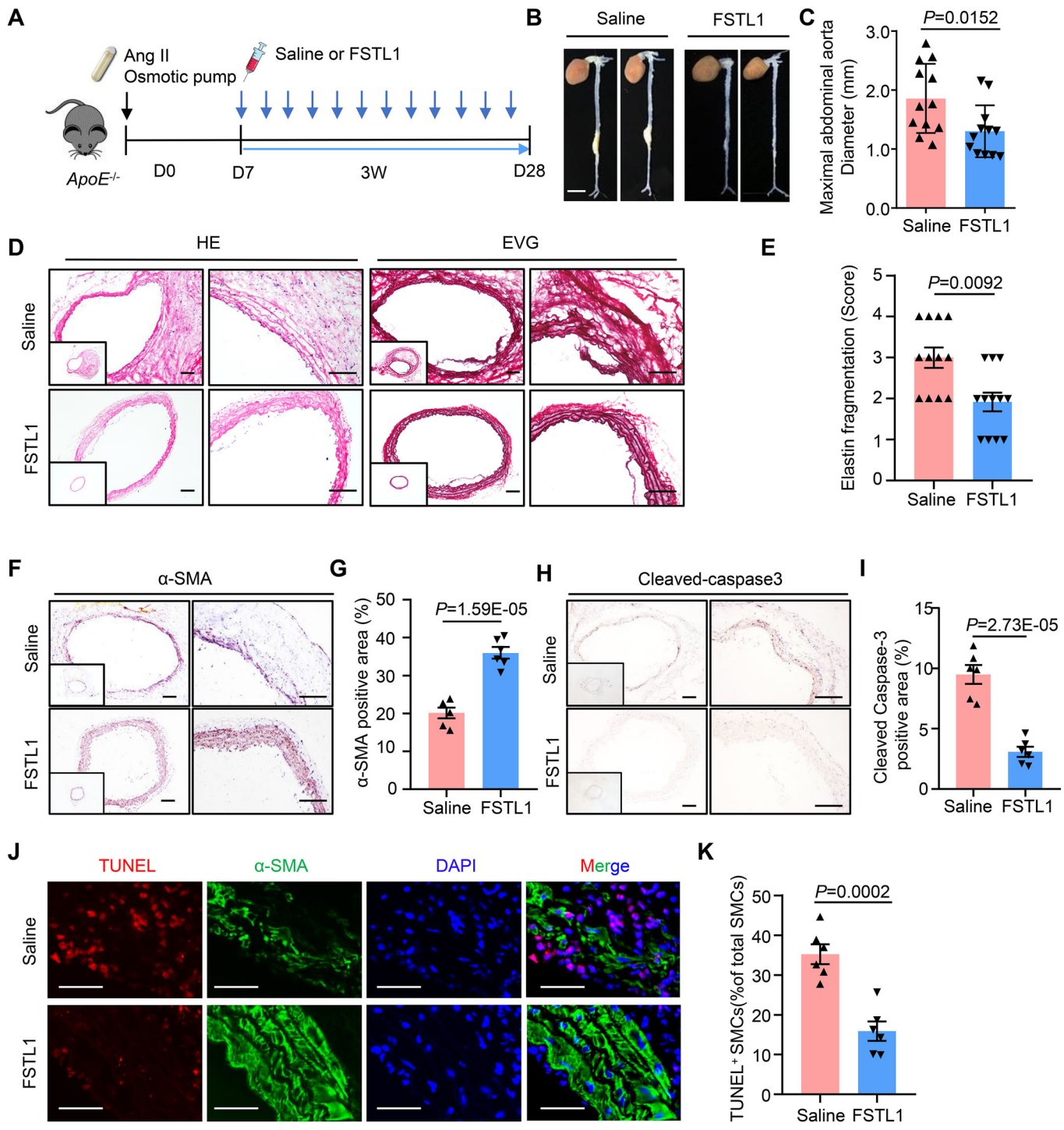

**Figure 8. Systemic FSTL infusion attenuates AAA development.**

(A) Experimental design. FSTL1 was injected intraperitoneally into Ang II-infused ApoE$^{-/-}$ male mice every other day for 3 weeks, starting on day 7 of AAA induction. (B) Representative photographs of the abdominal aorta (scale bar, 4 mm). (C) Quantification of maximal abdominal aortic diameters ($n = 12$). (D) Representative H&E and EVG staining images of abdominal aortic sections (scale bar, 50 μm). (E) Quantification of elastin fragmentation ($n = 12$). (F) Representative immunohistochemical staining of α-SMA in abdominal aortic sections (scale bar, 50 μm). (G) Quantification of α-SMA positive area as in (F) ($n = 6$). (H) Representative immunohistochemical staining of cleaved-Caspase 3 in abdominal aortic sections. (I) Quantification of cleaved-Caspase 3-positive area as in (H) ($n = 6$). (J) Representative TUNEL and α-SMA staining of abdominal aortic sections (scale bar, 25 μm). (K) Quantification of TUNEL-positive SMCs ($n = 6$). Student's $t$ test and Mann–Whitney $U$ tests were used to determine statistical difference. Source data are available online for this figure.

**Table 1.** Baseline characteristics of patients with and without AAA for tissue sampling.

| Variables | Without AAA ($n = 6$) | With AAA ($n = 6$) | P value |
|---|---|---|---|
| Men (%) | 5 (83.3) | 5 (83.3) | 0.999 |
| Age (years) | 42.0 ± 10.12 | 63.7 ± 8.2 | 0.004 |
| Smoker (%) | 1 (16.7) | 4 (66.7) | 0.242 |
| Drinker (%) | 1 (16.7) | 2 (33.3) | 0.999 |
| Body mass index, kg/m$^2$ | 21.7 ± 4.3 | 22.3 ± 2.2 | 0.700 |
| Systolic blood pressure, mmHg | 120.5 ± 24.4 | 152.2 ± 29.0 | 0.093 |
| Diastolic blood pressure, mmHg | 72.7 ± 22.9 | 81.5 ± 15.6 | 0.300 |
| LDL, mmol/L | 2.3 ± 0.4 | 3.1 ± 0.9 | 0.075 |
| HDL, mmol/L | 1.3 ± 0.6 | 1.0 ± 0.2 | 0.185 |
| Heart rate, bpm (%) | 88.5 ± 28.4 | 77.8 ± 17.8 | 0.589 |
| Hypertension (%) | 1 (16.7) | 5 (83.3) | 0.080 |
| Diabetes mellitus (%) | 1 (16.7) | 2 (33.2) | 0.999 |

*AAA* abdominal aortic aneurysm

**Table 2.** Baseline characteristics of patients with and without AAA for serum sampling.

| Variables | Without AAA ($n = 68$) | With AAA ($n = 55$) | P value |
|---|---|---|---|
| Men (%) | 61 (89.7%) | 49 (87.5%) | 0.919 |
| Age (years) | 69.4 (5.46) | 70.3 (5.18) | 0.346 |
| Smoker (%) | 31 (45.6%) | 26 (46.4%) | 1.000 |
| Drinker (%) | 14 (20.6%) | 15 (26.8%) | 0.550 |
| Body mass index, kg/m$^2$ | 23.7 (4.60) | 24.0 (2.85) | 0.608 |
| Systolic blood pressure, mmHg | 128 (21.0) | 138 (21.0) | 0.008 |
| Diastolic blood pressure, mmHg | 76.4 (12.0) | 78.4 (10.7) | 0.318 |
| LDL, mmol/L | 2.72 (0.93) | 2.81 (1.17) | 0.649 |
| HDL, mmol/L | 1.07 (0.36) | 0.99 (0.32) | 0.164 |
| Heart rate, bpm (%) | 80.0 (16.4) | 76.8 (13.2) | 0.236 |
| Hypertension (%) | 33 (48.5%) | 48 (85.7%) | <0.001 |
| Diabetes mellitus (%) | 20 (29.4%) | 21 (37.5%) | 0.447 |
| FSTL1, ng/mL | 41.3 (21.7) | 26.1 (11.2) | <0.001 |

*AAA* abdominal aortic aneurysm

100% TCA and shaken by vortexing. After incubation on ice for 30 min, centrifuge at 12,000 rpm at 4 °C for 15 min. Remove the supernatant and retain the sediment at the bottom. After the precipitation was cleaned with acetone several times, the bottom precipitation was retained by centrifugation and re-suspended in the sample buffer for subsequent western blot analysis.

## Western blot analysis

Equal amounts of proteins from each group were separated by SDS-PAGE and transferred to PVDF membranes. The membranes were incubated with primary antibodies at 4 °C overnight and HRP-labeled secondary antibodies for 1 h. Labeled proteins were visualized with an enhanced chemiluminescence system and quantified using the ImageJ software.

## Annexin V/propidium iodide (PI) assay

Cell apoptosis was evaluated by using an annexin V-FITC/PI staining Kit. Briefly, VSMCs were digested with 0.25% trypsin, then centrifuged at 1000 rpm for 5 min at 4 °C. Add 300 µl binding buffer into each flow tube and gently shake the tube to form a cell suspension. Add 5 µl annexin V-FITC and 5 µl propidium iodide into each tube and incubate at room temperature for 20 min. The samples were examined immediately on a flow cytometer. Data were analyzed using FlowJo software.

## RNA sequencing and DEGs analysis

Aortic tissues from Ang II-infused mice treated with CL316,243 or PBS, and the VSMCs treated with or without recombinant FSTL1 were used for RNA sequencing. Briefly, total RNA was extracted using Trizol reagent kit according to the manufacturer's protocol. RNA quality was assessed on an Agilent 2100 Bioanalyzer and checked using RNase-free agarose gel electrophoresis. After total RNA was extracted, eukaryotic mRNA was enriched by Oligo(dT) beads. Then the enriched mRNA was fragmented into short fragments using the fragmentation buffer and reverse transcribed

into cDNA by using NEB Next Ultra RNA Library Prep Kit for Illumina. The purified double-stranded cDNA fragments were end-repaired, A base added, and ligated to Illumina sequencing adapters. The ligation reaction was purified with the AMPure XP Beads (1.0X) and amplified by polymerase chain reaction. The resulting cDNA library was sequenced using Illumina Novaseq6000 by Gene Denovo Biotechnology Co. Differential expression analysis was performed by DESeq2 software. The genes/transcripts with the parameter of false discovery rate below 0.05 and absolute fold change ≥2 were considered differentially expressed genes.

## Sample preparation for secretome

The supernatant of the culture medium of adipocytes was collected and concentrated into dry powder under vacuum at −80 °C. 1 mg of the lyophilized powder was weighed and redissolved by adding 100 µL of 50 mmol/L $NH_4HCO_3$ solution, and the protein was precipitated by acetone overnight. Acetone and ethanol were used alternately to clean the protein precipitation. After centrifugation, the precipitation was freeze-dried and redissolved in urea. The protein was reduced by dithiothreitol for 60 min and alkylated by iodoacetamide for 45 min. After incubation with trypsin for 16 h, the reaction was terminated by trifluoroacetic acid solution. The peptides were eluted with 60% acetonitrile aqueous solution (containing 0.1% formic acid), 80% acetonitrile aqueous solution (containing 0.1% formic acid), and 100% acetonitrile (0.1% formic acid), and concentrated in a vacuum concentrator. The peptide concentration was tested using PierceTM Quantitative Colorimetric Peptide Assay, followed by LC-MS/MS analysis.

## LC-MS/MS analysis

Label-Free quantitative method was used to analyze the peptides from the cell culture media. The liquid phase was Easy1200 liquid system, packed with an integrated C18 Tip chromatographic analysis column (75 µm × 150 mm, 11.9 µm). Suitable mobile phase A: 0.1% formic acid/

**The paper explained**

**Problem**

The impact of adipose tissue on abdominal aortic aneurysm (AAA) development has received much attention, but whether brown remodeling of white adipose tissue would protect against AAA remains unclear.

**Results**

Patients with AAA had a decreased browning level of adipose tissue, and browning induction in mice attenuated AAA development. Browning adipocytes secreted a vessel-protective adipokine, Follistatin-like 1 (FSTL1), that inhibited vascular smooth muscle cell apoptosis. Adipocyte-specific FSTL1 deficiency abrogated the protective effect of browning induction, and supplementation of FSTL1 inhibited AAA development.

**Impact**

Our study suggests the therapeutic potential of adipose tissue browning and FSTL1 supplementation for treating AAA.

water solution; Mobile phase B: 0.1% formic acid, 80% acetonitrile/ aqueous solution, flow rate of 300 nL/min, gradient elution for 90 min, elution ratio is as follows: 0–2 min, 4–8% B phase; 2–72 min, 8–28% B phase; 72–82 min, 28–38% B phase; 82–85 min, 38–100% B phase; 85–90 min, 100% B phase. The Proteome Discoverer 2.4 software is used for qualitative statistical analysis of the original mass spectrum data. The software search parameters are set to: The maximum missing site was 3, the mass deviation of parent ion was ±10 ppm, searched using the UniProt protein database.

## Statistical analysis

No animal, sample, or data were excluded from the analysis. No blinding was performed. Animals were randomly grouped. The number of samples included in each experiment is included in the results section/figure legends. Continuous and categorical variables were analyzed with appropriate parametric or non-parametric tests based on distributional assumptions. Multivariable conditional logistic regression was performed, adjusting for age, sex, smoking status, drinking status, body mass index, systolic and diastolic blood pressure, heart rate, LDL, HDL, hypertension, and diabetes. Results were reported as odds ratios with 95% confidence intervals. Data are presented as the means ± SEM. All data were assessed for normality using the Shapiro–Wilk normality test. For normally distributed data, Student's $t$ test was used for comparing differences between groups, and one-way ANOVA or two-way ANOVA was employed for multiple group comparisons. For data not following a normal distribution, Mann–Whitney $U$ tests were used for two independent samples, and the Kruskal–Wallis H test was applied for multiple independent samples. Fisher's exact test was employed to analyze differences in percentages (e.g., AAA incidence). Statistical analyses were performed using GraphPad Prism.

## Data availability

RNA sequencing data are available at Gene Expression Omnibus (GEO) database number GSE296628 and GSE296762. The mass spectrometry proteomics data were available at ProteomeXchange Consortium database number PXD064197.

The source data of this paper are collected in the following database record: biostudies:S-SCDT-10_1038-S44321-025-00318-z.

## Peer review information

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

## Acknowledgements

We thank the Bioinformatics and Omics Center in Sun Yat-Sen Memorial Hospital for mass spectrometry analysis. This study was supported by grants from the National Natural Science Foundation of China (82170492, 82370445, U23A20397, 82271609, 82070237, 82270254, 82104433), Science and Technology Program of Guangzhou City of China (2023A03J0698), Guangdong Basic and Applied Basic Research Foundation (2022A1515111231, 2024A1515011192), the Science and Technology Planning Project of Guangdong Province (2023B1212060013).

## Author contributions

**Chunling Huang**: Conceptualization; Data curation; Software; Formal analysis; Validation; Investigation; Visualization; Methodology. **Yuna Huang**: Data curation; Software; Validation; Investigation; Visualization; Methodology. **Boshui Huang**: Conceptualization; Resources; Data curation; Formal analysis; Validation; Investigation; Visualization; Methodology. **Lei Yao**: Conceptualization; Data curation; Software; Formal analysis; Validation; Investigation; Visualization; Methodology. **Zenghui Zhang**: Investigation; Methodology. **Luoxiao Dong**: Investigation; Methodology. **Chang Guan**: Funding acquisition; Investigation; Methodology. **Junping Li**: Formal analysis; Validation. **Zhaoqi Huang**: Validation; Investigation; Methodology. **Sixu Chen**: Validation; Investigation. **Yuan Jiang**: Funding acquisition; Validation; Investigation. **Yuling Zhang**: Validation; Investigation. **Jingfeng Wang**: Supervision; Funding acquisition. **Yangxin Chen**: Conceptualization; Supervision; Funding acquisition. **Zhaoyu Liu**: Conceptualization; Supervision; Funding acquisition; Writing—original draft; Project administration; Writing—review and editing.

Source data underlying figure panels in this paper may have individual authorship assigned. Where available, figure panel/source data authorship is listed in the following database record: biostudies:S-SCDT-10_1038-S44321-025-00318-z.

## Disclosure and competing interests statement

The authors declare no competing interests.

# Expanded View Figures

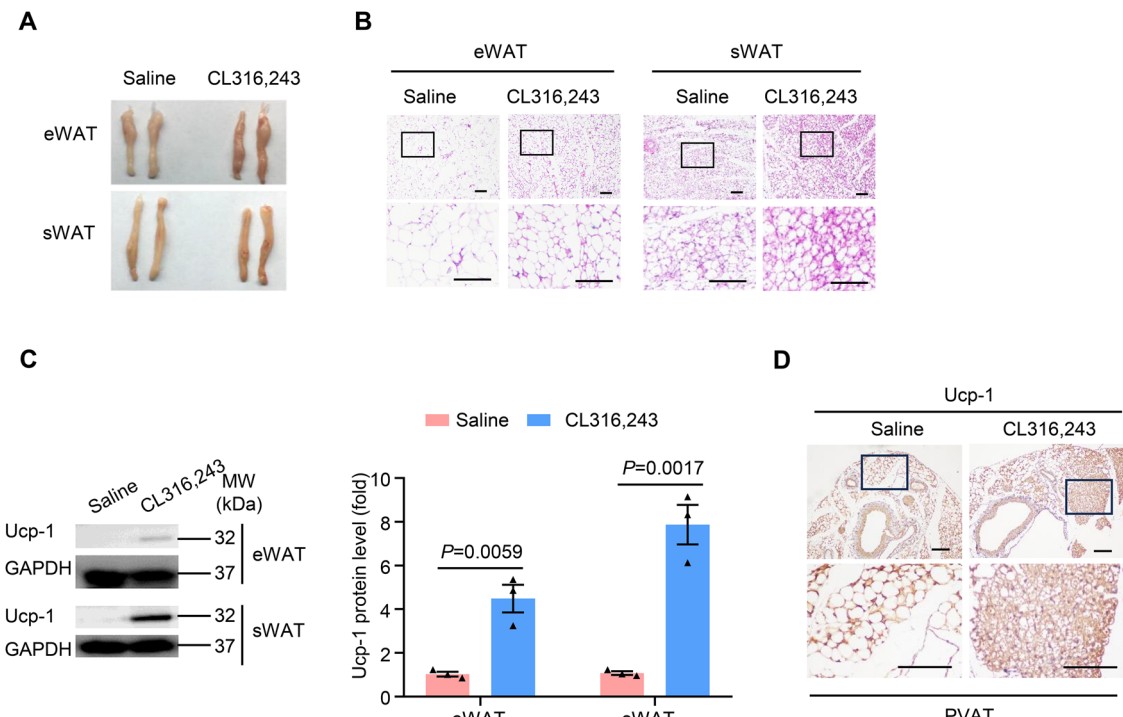

**Figure EV1.  CL316,243 administration successfully induced adipose tissue browning.**

(A) Gross appearance of epididymal white adipose tissue (eWAT)and subcutaneous white adipose tissue (sWAT). (B) H&E staining. (C) Western blot analysis of Ucp-1 expression. (D) Immunohistochemical staining of Ucp-1 in perivascular adipose tissue ($n = 3$). Scale bar, 200 μm. Student's *t* test was used to determine statistical difference.

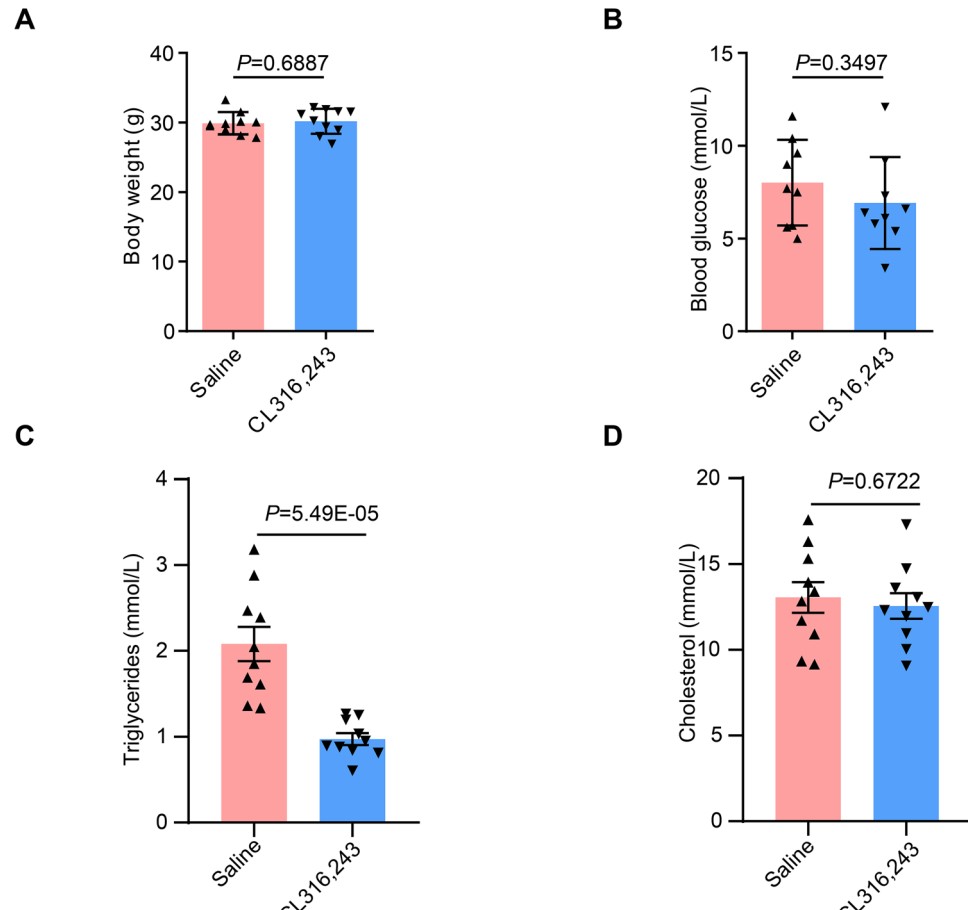

**Figure EV2.  Characteristics of ApoE$^{-/-}$ mice after CL316,243 treatment.**

(**A**) Body weight ($n = 10$). (**B**) Blood glucose levels ($n = 9$). (**C**) Serum triglycerides ($n = 10$). (**D**) Serum cholesterol ($n = 10$). Student's $t$ test was used to determine statistical difference.

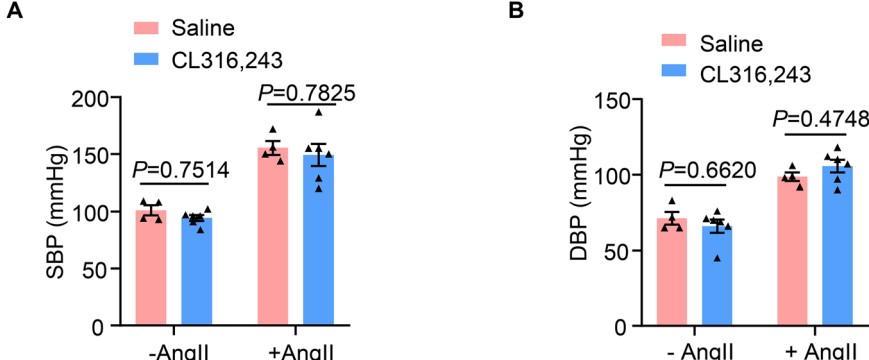

**Figure EV3. CL316,243 administration did not significantly change systolic or diastolic blood pressures.**

(A) Systolic blood pressure. (B) Diastolic blood pressure ($n = 4, 6$). Two-way ANOVA was used to determine statistical difference.

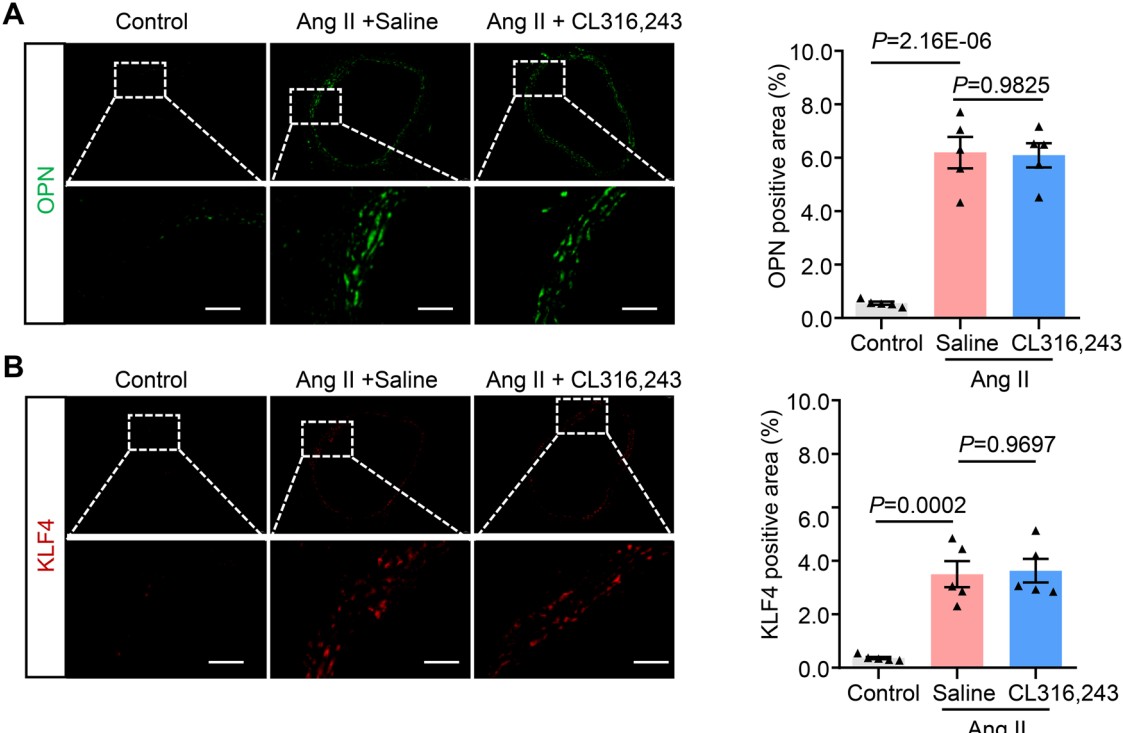

**Figure EV4.  CL316,243 administration did not alter the expression of VSMC phenotypic switch markers.**

Immunofluorescence staining of Osteopontin (OPN) (**A**) and krüppel-like factor 4 (KLF4) (**B**) in aortic tissues from mice treated with saline or CL316,243 infused with Ang II ($n = 5$; scale bar, 25 μm). One-way ANOVA was used to determine statistical difference.

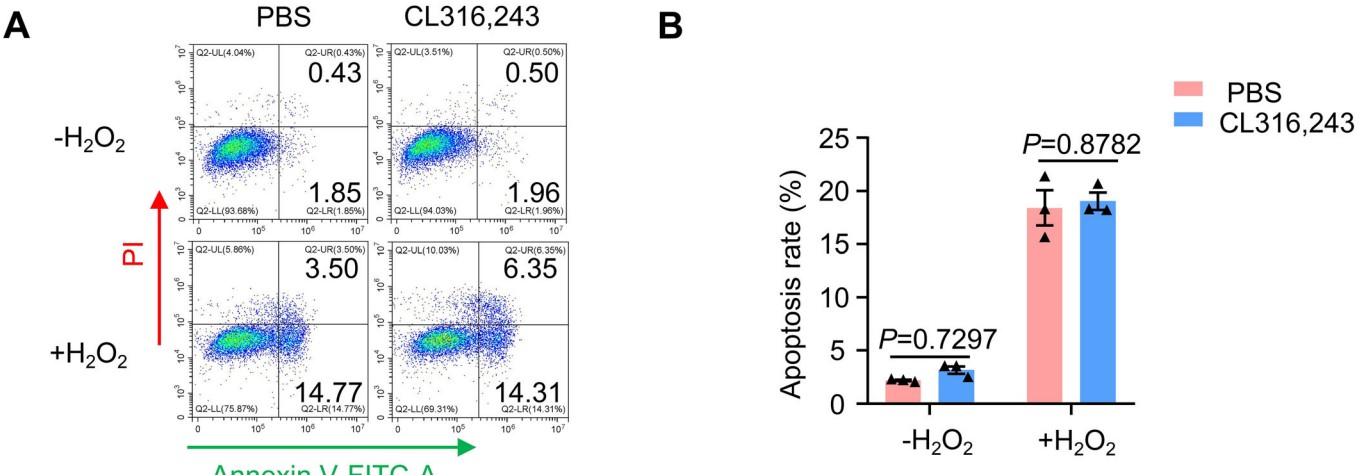

**Figure EV5. CL316,243 did not have a direct effect on VSMC apoptosis.**

(A) Flow cytometric analysis of Annexin V-FITC staining in VSMCs treated with PBS or CL316,243 and then stimulated with $H_2O_2$ for 24 h. (B) Quantification of apoptosis as in (A) ($n = 3$). Two-way ANOVA was used to determine statistical difference.

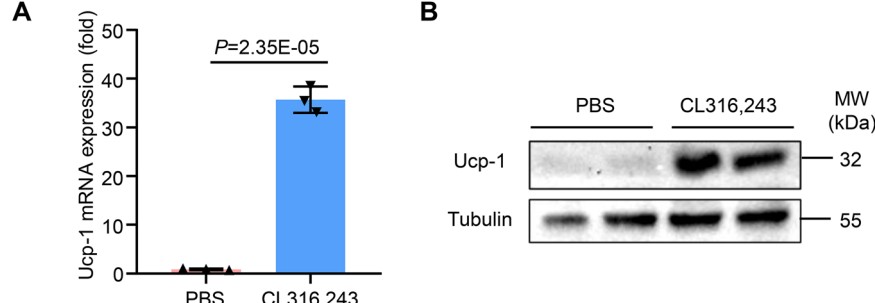

**A**

**B**

**Figure EV6. CL316,243 treatment robustly induced adipocyte browning.**

Primary adipocytes were treated with PBS or CL316,243 for 24 h. (**A**) Quantitative RT-PCR analysis of Ucp-1 mRNA level ($n = 3$). (**B**) Western blot analysis of Ucp-1 protein level. Student's $t$ test was used to determine statistical difference.

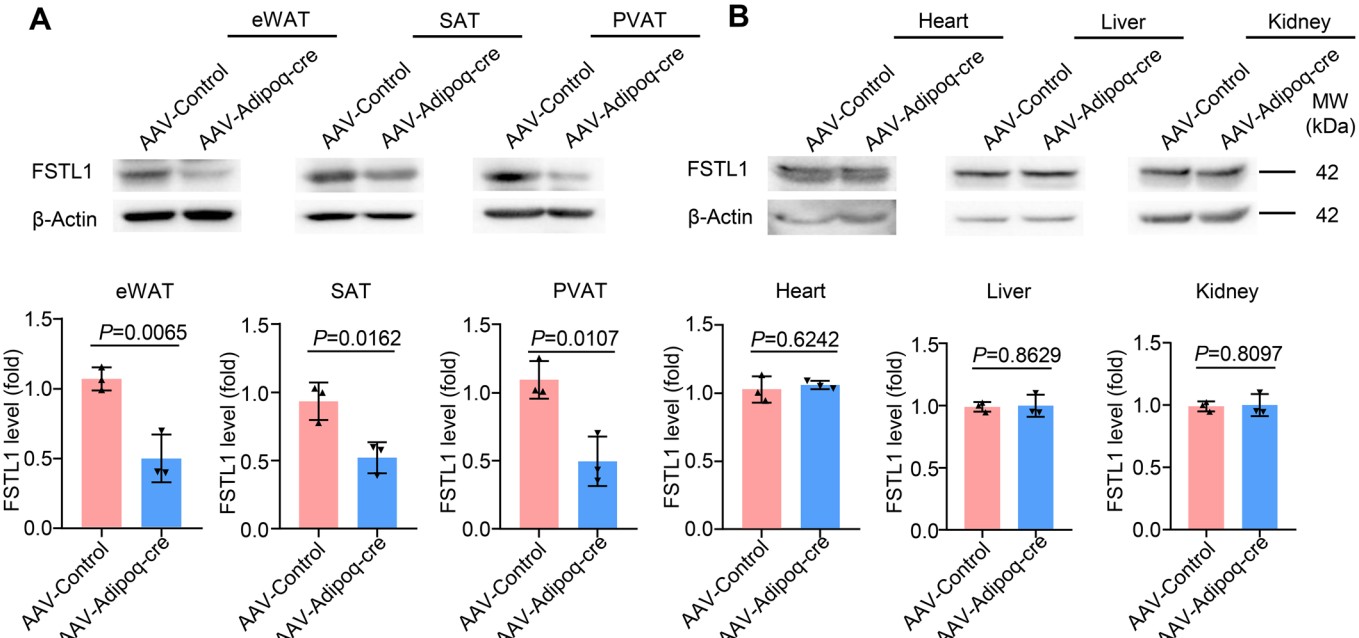

**Figure EV7. Western blot analysis of FSTL1 expression after AAV injection and CL316,243 treatment.**

(A) Expression of FSTL1 in adipose tissues. (B) Expression of FSTL1 in other tissues ($n = 3$). Student's *t* test was used to determine statistical difference.

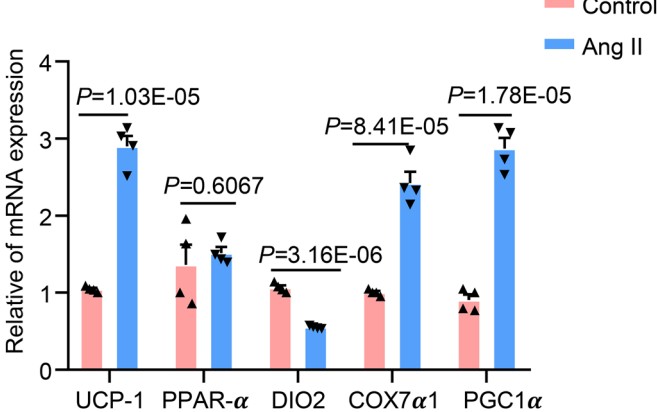

**Figure EV8. Quantitative PCR analysis of mRNA expression of browning associated genes in PVAT of ApoE−/− mice treated with control or Ang II (n = 4).**

Student's *t* test was used to determine statistical difference.

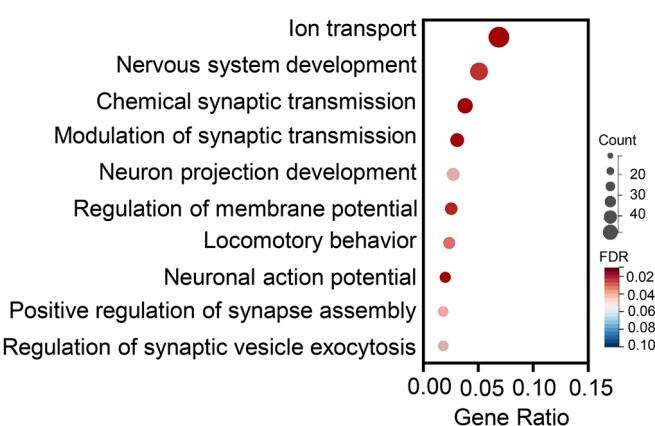

**Figure EV9.** Gene ontology (GO) analysis of downregulated proteins in aortic tissue from mice treated with saline or CL316,243 infused with Ang II.

