## [Peer Review File · EMBO Molecular Medicine]

Brown remodeling of white adipose tissue protects against abdominal aortic aneurysm via batokine FSTL1

Zhaoyu Liu, Chunling Huang, Yuna Huang, Boshui Huang, Lei Yao, Zenghui Zhang, Luoxiao Dong, Chang Guan, Junping Li, Zhaoqi Huang, Sixu Chen, Yuan Jiang, Yuling Zhang, Jingfeng Wang, and Yangxin Chen

Corresponding authors: Zhaoyu Liu (liuzhy98@mail.sysu.edu.cn) , Jingfeng Wang (wjingf@mail.sysu.edu.cn), Yangxin Chen (chenyx39@mail.sysu.edu.cn)

Review Timeline:

Submission Date:	20th Nov 24
Editorial Decision:	18th Dec 24
Revision Received:	1st Jun 25
Editorial Decision:	1st Jul 25
Revision Received:	29th Jul 25
Editorial Decision:	3rd Sep 25
Revision Received:	5th Sep 25
Accepted:	16th Sep 25

Editor: Lise Roth

Transaction Report:

18th Dec 2024

Dear Prof. Liu,

Thank you for the submission of your manuscript to EMBO Molecular Medicine. Unfortunately, one referee had to be withdrawn due to personal circumstances but given that the other two referees provide similar recommendations, we prefer to make a decision now in order to avoid further delay in the process.

As you will see from the reports below, both referees mention the novelty and potential translational interest of the work, however they also highlight several concerns that should be addressed, including (but not limited to) the choice of 2 mouse models, the lack of details in the methods, and the choice of in vitro cells.

Addressing these points and the other reviewers' concerns in full will be necessary for further considering the manuscript in our journal, and acceptance of the manuscript will entail a second round of review. EMBO Molecular Medicine encourages a single round of revision only and therefore, acceptance or rejection of the manuscript will depend on the completeness of your responses included in the next, final version of the manuscript. For this reason, and to save you from any frustrations in the end, I would strongly advise against returning an incomplete revision.

We are expecting your revised manuscript within three months, if you anticipate any delay, please contact us.

We require:

- 1) A .docx formatted version of the manuscript text (including legends for main figures, EV figures and tables). Please make sure that the changes are highlighted to be clearly visible.
- 2) Individual production quality figure files as .eps, .tif, .jpg (one file per figure). For guidance, download the 'Figure Guide PDF' (<https://www.embopress.org/page/journal/17574684/authorguide#figureformat>).
- 3) At EMBO Press we ask authors to provide source data for the main figures. Our source data coordinator will contact you to discuss which figure panels we would need source data for and will also provide you with helpful tips on how to upload and organize the files.
- 4) A .docx formatted letter INCLUDING the reviewers' reports and your detailed point-by-point responses to their comments. As part of the EMBO Press transparent editorial process, the point-by-point response is part of the Review Process File (RPF), which will be published alongside your paper.
- 5) A complete author checklist, which you can download from our author guidelines (<https://www.embopress.org/page/journal/17574684/authorguide#submissionofrevisions>). Please insert information in the checklist that is also reflected in the manuscript. The completed author checklist will also be part of the RPF.
- 6) All Materials and Methods need to be described in the main text using our 'Structured Methods' format. According to this format, the Methods section includes a Reagents and Tools Table (listing key reagents, experimental models, software and relevant equipment and including their sources and relevant identifiers) followed by a Methods and Protocols section describing the methods, ideally using a step-by-step protocol format. The aim is to facilitate adoption of the methodologies across labs. Please download and fill our Reagents and Tools Table template (.docx), which you can find in our author guidelines: <https://www.embopress.org/page/journal/14693178/authorguide#structuredmethods>. When submitting your revised manuscript, please do not include the Reagents and Tools Table in the Methods section of the manuscript but upload it as a separate file choosing the file type "Reagent Table". An example of a Method paper with Structured Methods can be found here: <https://www.embopress.org/doi/10.15252/msb.20178071>
- 7) Please note that all corresponding authors are required to supply an ORCID ID for their name upon submission of a revised manuscript.
- 8) It is mandatory to include a 'Data Availability' section after the Materials and Methods. Before submitting your revision, primary datasets produced in this study need to be deposited in an appropriate public database, and the accession numbers and

database listed under 'Data Availability'. Please remember to provide a reviewer password if the datasets are not yet public (see <https://www.embopress.org/page/journal/17574684/authorguide#dataavailability>).

9) For data quantification: please specify the name of the statistical test used to generate error bars and P values, the number (n) of independent experiments (specify technical or biological replicates) underlying each data point and the test used to calculate p-values in each figure legend. The figure legends should contain a basic description of n, P and the test applied. Graphs must include a description of the bars and the error bars (s.d., s.e.m.). Please provide exact p values.

10) Our journal encourages inclusion of *data citations in the reference list* to directly cite datasets that were re-used and obtained from public databases. Data citations in the article text are distinct from normal bibliographical citations and should directly link to the database records from which the data can be accessed. In the main text, data citations are formatted as follows: "Data ref: Smith et al, 2001" or "Data ref: NCBI Sequence Read Archive PRJNA342805, 2017". In the Reference list, data citations must be labeled with "[DATASET]". A data reference must provide the database name, accession number/identifiers and a resolvable link to the landing page from which the data can be accessed at the end of the reference. Further instructions are available at .

11) We replaced Supplementary Information with Expanded View (EV) Figures and Tables that are collapsible/expandable online. A maximum of 5 EV Figures can be typeset. EV Figures should be cited as 'Figure EV1, Figure EV2' etc... in the text and their respective legends should be included in the main text after the legends of regular figures.

12) The paper explained: EMBO Molecular Medicine articles are accompanied by a summary of the articles to emphasize the major findings in the paper and their medical implications for the non-specialist reader. Please provide a draft summary of your article highlighting

13) Author contributions: CRediT has replaced the traditional author contributions section because it offers a systematic machine readable author contributions format that allows for more effective research assessment. Please remove the Authors Contributions from the manuscript and use the free text boxes beneath each contributing author's name in our system to add specific details on the author's contribution. More information is available in our guide to authors.

Please also suggest a visual abstract to illustrate your article as a PNG file 550 px wide x 300-600 px high. A cropped portion of this image will serve as thumbnail for the table of content on our webpage.

16) As part of the EMBO Publications transparent editorial process initiative (see our Editorial at <http://embomolmed.embopress.org/content/2/9/329>), EMBO Molecular Medicine will publish online a Review Process File (RPF) to accompany accepted manuscripts.

In the event of acceptance, this file will be published in conjunction with your paper and will include the anonymous referee reports, your point-by-point response and all pertinent correspondence relating to the manuscript. Let us know whether you

agree with the publication of the RPF and as here, if you want to remove or not any figures from it prior to publication. Please note that the Authors checklist will be published at the end of the RPF.

I look forward to receiving your revised manuscript.

Yours sincerely,

Lise Roth

**** Reviewer's comments ****

Referee #2 (Comments on Novelty/Model System for Author):

Overall, the study is of interest and the sequence of experiments performed support the conclusions. There are some technical issues that should be improved (quantifications, photographs) and some additional experiments that would enhance the translational perspective of the results mainly observed in experimental models.

In addition, the authors included information from a previous dataset in Fig 1C-F. We understand that the information is available and they only performed a reanalysis but we do not know if this should be acknowledged or commented in the methods/results section (it only appears in the figure legend)

Referee #2 (Remarks for Author):

The paper by Huang et al tested how browning adipose tissue could impact abdominal aortic aneurysm (AAA) in both human samples and experimental models. The authors used several methodological approaches to identify the potential mediator and mechanisms underlying the protective effect of browning adipose tissue in experimental AAA. Finally, they performed a therapeutic strategy to supply the main mediator identified (FSTL1) in an experimental model of AAA. Overall, the study is original and the results are interesting. However, some issues must be improved.

Main concerns

- Human samples: UCP-1 must be quantified by more quantitative techniques (Q-PCR, western blot). Among the clinical characteristics, BMI should be included. The analysis of FSTL1 in a large cohort of tissue and/or plasma of controls and AAA patients would increase the potential translation of the data to human disease.
- AngII model: The authors should include mRNA analysis of browning associated genes described in the human samples. Photographs are overexposed and it is not clear how quantification of different markers have been performed (eg. SMA is presented as relative fluorescence intensity instead of % of SMA positivity normalized by total area). Analysis of other markers of SMC phenotype switch should be tested.
- In vitro studies: The reason why studies in human vascular SMCs instead of primary culture mouse VSMCs are not clear since the authors used mouse adipocytes as the source of secreted proteins. In addition, the use of elastase as a more pathological AAA mediator to induce VSMC apoptosis would be of interest
- CaPO4 model: there is no explanation to change the initial AngII model of AAA, even there are known differences between this CaPO4 model and that of AngII. The % of SMA should also be tested in this model.

Minor concerns

- The analysis of SMA in the figure 8 is performed in less animals than the analysis of elastin content. Please explain why

- .- The references of the antibodies used should be included. In this respect, we guess the antibody used for caspase 3 (CST) is Cleaved Caspase-3 (Asp175) Antibody #9661 from Cell Signalling Technology, and then the authors should test total caspase 3 levels (#9662 from Cell Signalling Technology) .- There are some asterisks inside the figures (eg. Check figure 4F)
- .- Line 49: "Several clinical studies...6-8." This statement includes experimental studies, please revise.
- .-Fig 2B. There should be some interesting downregulated proteins, so they should also be included.
- .- Fig S2.-Serum lipid analysis should be also analyzed in this model (<https://doi.org/10.1016/j.phrs.2021.105524>)
- .- The analysis of glucose were performed in fasting animals? Diabetes 1997 Aug;46(8):1257-63. doi: 10.2337/diab.46.8.1257.

Referee #3 (Comments on Novelty/Model System for Author):

The manuscript "Brown remodeling of white adipose tissue protects against abdominal aortic aneurysm via a novel batokine FSTL1" (Chunling Huang et al, #EMM-2024-20948) describes the effect of browning of the white adipose tissue on abdominal aortic aneurysm (AAA) in two mouse models of AAA. The authors employed RNA sequencing and proteomic analysis to identify Follistatin-like 1 (FSTL1) secreted during browning as a protective factor. The mechanism and signaling of this protection were addressed, and it was found that FSTL1 inhibited VSMC apoptosis through DIP2A/AKT. The effect of FSTL1 was confirmed in adipocyte-specific deficiency of the adipokine that resulted in a loss of the effect of browning on AAA (infection of Flox-FSTL1 mice with adenovirus carrying adipocyte-specific Cre). Supplementation of FSTL1 in AAA model, where there is a decrease of browning, protected from AAA. Importantly, the study was conducted in two mouse models and in cultured VSMC and includes human tissue samples of perivascular adipose tissue from AAA patients and control tissue from patients without AAA. The results are consistent across the models.

The manuscript is written in a clear logical manner and offers a clear representation of the results that are novel and assign a new vascular function to FSTL1. Evaluation of the effects is thorough and includes multiple browning markers, two AAA models, and several indexes of AAA and browning. The effect of systemic injection after AAA development is impressive and demonstrates a therapeutic potential.

Overall, this is a complete and well-designed and well-executed study with important conclusions and mechanistic insights. Lack of some important methodological details complicates the evaluation of the results quality. Methods section is incomplete and should be revised as suggested below, with detailed description of all approaches used to generate the results.

1. I am not sure why FSTL1 is called "novel" - there is quite large body of literature on this protein, including publications describing effects in cardiovascular system, secretion by AT, and effects on VSMC. One report from 2012 shows that FSTL1 levels predict development of AAA (PMID: 22316625). Sadly, these publications have not been cited. The authors should revise the references to add all these publications, including a couple of recent reports from 2024.
2. Evaluation of browning should be described in Methods or in supplement, with references to original reports describing the markers.
3. Quantification of staining should be described in Methods or Supplement.
4. Why were two mouse AAA models used? This should be explained and discussed.
5. The selection of FST1 out of 18 proteins that were upregulated in Fig.4(A-D) has not been discussed or justified. Were other proteins also tested? Are they involved as well? There are several that are known to effect VSMC functions. These results should be also discussed in the Discussion section.
6. In Figure 5C and D, it does not seem that representative blots were used: while the quantification shows no decrease or even increase of PARP in si-DIP2A-treated cells in response to FSTL1, there is an obvious decrease in the blots.
7. Figure legends should be revised to include explanations of abbreviated labels in figures (including Supplemental Figures), even if abbreviations are spelled out in the text.
8. The discussion largely repeats the introduction - these sections should be reconciled. The discussion sections should be used to discuss the current study and results, implications and limitations etc.
9. Evaluation of elastin fragmentation should be better described in Methods (Supplement). What were criteria for no elastin degradation, mild degradation, moderate, moderate to severe, severe elastin degradation? How was this quantified?

Minor:

- a) Please correct the second sentence in the abstract, "adipose tissue" used twice: "Emerging evidences have suggested a crosstalk between adipose tissue and vascular cells and brown adipose tissue is beneficial for cardiovascular health." "Evidence" should be in singular form, "emerging evidence".
- b) Spellcheck is recommended - there are typos, e.g., "decreased" is misspelled several times as "deceased", there are other typos in the text.

Referee #3 (Remarks for Author):

The manuscript "Brown remodeling of white adipose tissue protects against abdominal aortic aneurysm via a novel batokine

FSTL1" (Chunling Huang et al, #EMM-2024-20948) describes the effect of browning of the white adipose tissue on abdominal aortic aneurism (AAA) in two mouse models of AAA. The authors employed RNA sequencing and proteomic analysis to identify Follistatin-like 1 (FSTL1) secreted during browning as a protective factor. The mechanism and signaling of this protection were addressed, and it was found that FSTL1 inhibited VSMC apoptosis through DIP2A/AKT. The effect of FSTL1 was confirmed in adipocyte-specific deficiency of the adipokine that resulted in a loss of the effect of browning on AAA (infection of Flox-FSTL1 mice with adenovirus carrying adipocyte-specific Cre). Supplementation of FSTL1 in AAA model, where there is a decrease of browning, protected from AAA. Importantly, the study was conducted in two mouse models and in cultured VSMC and includes human tissue samples of perivascular adipose tissue from AAA patients and control tissue from patients without AAA. The results are consistent across the models.

The manuscript is written in a clear logical manner and offers a clear representation of the results that are novel and assign a new vascular function to FSTL1. Evaluation of the effects is thorough and includes multiple browning markers, two AAA models, and several indexes of AAA and browning. The effect of systemic injection after AAA development is impressive and demonstrates a therapeutic potential.

Overall, this is a complete and well-designed and well-executed study with important conclusions and mechanistic insights. Lack of some important methodological details complicates the evaluation of the results quality. Methods section is incomplete and should be revised as suggested below, with detailed description of all approaches used to generate the results.

1. I am not sure why FSTL1 is called "novel" - there is quite large body of literature on this protein, including publications describing effects in cardiovascular system, secretion by AT, and effects on VSMC. One report from 2012 shows that FSTL1 levels predict development of AAA (PMID: 22316625). Sadly, these publications have not been cited. The authors should revise the references to add all these publications, including a couple of recent reports from 2024.
2. Evaluation of browning should be described in Methods or in supplement, with references to original reports describing the markers.
3. Quantification of staining should be described in Methods or Supplement.
4. Why were two mouse AAA models used? This should be explained and discussed.
5. The selection of FST1 out of 18 proteins that were upregulated in Fig.4(A-D) has not been discussed or justified. Were other proteins also tested? Are they involved as well? There are several that are known to effect VSMC functions. These results should be also discussed in the Discussion section.
6. In Figure 5C and D, it does not seem that representative blots were used: while the quantification shows no decrease or even increase of PARP in si-DIP2A-treated cells in response to FSTL1, there is an obvious decrease in the blots.
7. Figure legends should be revised to include explanations of abbreviated labels in figures (including Supplemental Figures), even if abbreviations are spelled out in the text.
8. The discussion largely repeats the introduction - these sections should be reconciled. The discussion sections should be used to discuss the current study and results, implications and limitations etc.
9. Evaluation of elastin fragmentation should be better described in Methods (Supplement). What were criteria for no elastin degradation, mild degradation, moderate, moderate to severe, severe elastin degradation? How was this quantified?

Minor:

- a) Please correct the second sentence in the abstract, "adipose tissue" used twice: "Emerging evidences have suggested a crosstalk between adipose tissue and vascular cells and brown adipose tissue is beneficial for cardiovascular health." "Evidence" should be in singular form, "emerging evidence".
- b) Spellcheck is recommended - there are typos, e.g., "decreased" is misspelled several times as "deceased", there are other typos in the text.

RESPONSE TO EDITOR AND REVIEWERS

We sincerely thank the editor and reviewers for their constructive comments. We have carefully revised our manuscripts, incorporating your valuable feedback. Below are our point-to-point responses, and all the changes in the revised manuscript are highlighted in red.

EDITOR COMMENTS

As you will see from the reports below, both referees mention the novelty and potential translational interest of the work, however they also highlight several concerns that should be addressed, including (but not limited to) the choice of 2 mouse models, the lack of details in the methods, and the choice of in vitro cells.

***Response:** Thank you and both referees for the very positive comments about the novelty and potential translational interest of our work. We have explained and discussed the two mouse models and the choice of in vitro cells. Besides, we have added more details in the methods and carefully addressed other concerns of the reviewers as in the specific responses below.*

REVIEWER COMMENTS

Referee #2 (Comments on Novelty/Model System for Author):

Overall, the study is of interest and the sequence of experiments performed support the conclusions. There are some technical issues that should be improved (quantifications, photographs) and some additional experiments that would enhance the translational perspective of the results mainly observed in experimental models.

***Response:** Thank you for the positive comments and valuable suggestions. We have addressed the mentioned technical issues, including the quantifications and photograph images. In addition, we have performed additional clinical sample detection to enhance the translational significance as suggested. The data will be shown below.*

In addition, the authors included information from a previous dataset in Fig 1C-F. We understand that the information is available and they only performed a reanalysis but we do not know if this should be acknowledged or commented in the methods/results section (it only appears in the figure legend)

***Response:** Thanks for the thoughtful reminder. We have added acknowledgement of the GEO dataset in both results and methods section in the revised manuscript.*

Referee #2 (Remarks for Author):

The paper by Huang et al tested how browning adipose tissue could impact abdominal aortic aneurysm (AAA) in both human samples and experimental models. The authors used several methodological approaches to identify the potential mediator and mechanisms underlying the protective effect of browning adipose tissue in experimental AAA. Finally, they performed a therapeutic strategy to supply the main mediator identified (FSTL1) in an experimental model of AAA. *Overall, the study is original and the results are interesting.* However, some issues must be improved.

Response: Thank you for your professional summary and positive comments. We have revised our manuscript according to your constructive suggestions.

Main concerns

- Human samples: UCP-1 must be quantified by more quantitative techniques (Q-PCR, western blot). Among the clinical characteristics, BMI should be included.

Response: Thank you for the good suggestions. We have quantified UCP-1 levels in perivascular adipose tissue in human samples by Q-PCR and western blot. The results showed that UCP-1 was significantly downregulated in adipose tissues from AAA patients. These data have been added into Figure 1 as Fig 1C and Fig 1D in the revised manuscript.

Figure 1. C, Q-PCR analysis of UCP-1 mRNA level in PVAT of AAA patients and control. D, Western blot analysis of UCP-1 protein level in PVAT of AAA patients and control (n=6).

Besides, BMI was included into the clinical characteristics in Table EV1 (previously Table S1) in the revised manuscript.

Table EV1. Baseline characteristics of patients with and without AAA

Variables	Without AAA (n=6)	With AAA (n=6)	P value
Men (%)	5 (83.3)	5 (83.3)	0.999
Age (years)	42.0±10.12	63.7 ± 8.2	0.004
Smoker (%)	1 (16.7)	4 (66.7)	0.242
Drinker (%)	1 (16.7)	2 (33.3)	0.999
Body mass index, kg/m ²	21.7±4.3	22.3±2.2	0.700
Systolic blood pressure, mmHg (%)	120.5±24.4	152.2±29.0	0.093
Diastolic blood pressure, mmHg (%)	72.7 ±22.9	81.5 ± 15.6	0.300
LDL (%)	2.3±0.4	3.1 ± 0.9	0.075
HDL (%)	1.3±0.6	1.0±0.2	0.185
Heart rate, bpm (%)	88.5 ±28.4	77.8 ± 17.8	0.589
Hypertension (%)	1 (16.7)	5 (83.3)	0.080
Diabetes mellitus (%)	1 (16.7)	2 (33.2)	0.999

Note: AAA, abdominal aortic aneurysm

The analysis of FSTL1 in a large cohort of tissue and/or plasma of controls and AAA patients would increase the potential translation of the data to human disease.

Response: Thank you for the good suggestion. We have analyzed FSTL1 levels in the plasma of control (n=68) and AAA patients(n=55). As shown below, FSTL1 levels were significantly decreased in the plasma of AAA patients. We have added this data in the revised manuscript as Figure 4G.

Figure 4G, ELISA analysis of FSTL1 in the plasma of control (n=68) and AAA patients(n=55).

-AngII model: The authors should include mRNA analysis of browning associated genes described in the human samples. Photographs are overexposed and it is not clear how quantification of different markers have been performed (eg. SMA is presented as relative fluorescence intensity instead of % of SMA positivity normalized by total area). Analysis of other markers of SMC phenotype switch should be tested

Response: Thank you for the suggestion. We have detected the browning associated genes in mouse Ang II models. We found that Ang II stimulation increased the expression of those genes in the adipose tissues (data shown below). This is consistent with previous studies (PMID: 29263921, PMID: 35799891). Since Ang II has been known to induce inflammatory response which requires increased energy expenditure, this increase in browning may reflect a compensatory response to the heightened inflammation and metabolic stress induced by Ang II. As AAA progresses, chronic inflammation and metabolic dysfunction may override this protective mechanism, leading to a decline in browning levels.

Figure. Q-PCR analysis of mRNA expression of browning associated genes in PVAT of ApoE-/- mice treated with control or AngII (n=4).

For the overexposed images, we have replaced them with better ones (Figure 2C and Figure 2G). For the quantification, we have calculated the fluorescence and DAB intensity normalized by aortic wall area as suggested in the revised manuscript (Figure 1D, 1F, 8G and 8I).

In addition, we have performed the immunofluorescence staining of some other markers of SMC phenotype switch such as Krüppel-like factor 4 (KLF4) and osteopontin (OPN) in the aortic tissues. The results showed that AngII significantly increased the level of KLF4 and OPN, whereas additional CL316,243 administration did not change their levels. The data are shown below.

Figure. Immunofluorescence staining of Krüppel-like factor 4 (KLF4) and osteopontin (OPN) in the aortic tissues (n=5).

- In vitro studies: The reason why studies in human vascular SMCs instead of primary culture mouse VSMCs are not clear since the authors used mouse adipocytes as the source of secreted proteins. In addition, the use of elastase as a more pathological AAA mediator to induce VSMC apoptosis would be of interest.

Response: Thanks for the insightful questions. We agree that using mouse VSMCs would be more appropriate if the source of secreted proteins is mouse adipocytes, which maintains species-specific signaling compatibility. The main reason why we use human VSMCs instead of primary culture mouse VSMCs is that it is difficult to get enough mouse VSMCs to meet the experimental requirement because the amount of VSMCs obtained from each mouse is very limited and those cells are also very difficult to maintain a good state after passage. Considering the core apoptotic signaling pathways are conserved between human and mouse VSMCs, and human VSMCs have its advantage for translational studies due to their direct clinical applicability, we used human VSMCs instead of primary culture mouse VSMCs.

However, future studies are warranted to validate these findings using either mouse adipocyte-mouse VSMC or human adipocyte-human VSMC coculture systems. We have added the acknowledgement of this limitation in the revised discussion.

For the use of elastase, yes, elastase can act as a pathological mediator in AAA and can contribute to VSMC apoptosis *in vivo*. However, the mechanism of elastase is to function as a protease that degrades extracellular matrix (ECM) components like elastin and collagen in the aortic wall, and ECM degradation disrupts VSMC-ECM interactions, which leads to VSMC apoptosis. Therefore, the effect of elastase on VSMC apoptosis is not directly, but context-dependently. Thus, in the current study, we did not use elastase, but rather TNF α +CHX and H₂O₂ to induce VSMC apoptosis *in vitro*, both of which are widely used in the field to mimic the direct inflammatory and oxidative stimuli in AAA (Circulation, 2020, PMID: 32354235; JCI, 2022, PMID: 36066968; JCI Insight, 2023, PMID: 37079380)

- CaPO4 model: there is no explanation to change the initial AngII model of AAA, even there are known differences between this CaPO4 model and that of AngII. The % of SMA should also be tested in this model.

Response: Thank you for the question and good suggestions. Ang II model and CaPO4 model are commonly used mouse models for studying AAA. The main reason we changed the initial Ang II model to CaPO4 model was due to a technical issue. The Ang II infusion model results in the formation of aortic aneurysms at a variable location, typically above the renal artery bifurcation, which site is not conducive to our later surgical procedures aimed at wrapping hydrogels around the abdominal aorta. In contrast, the CaPO4 incubation model allows for the operation to be performed at a fixed location in the infrarenal region, which facilitates hydrogel wrapping for local protein delivery. That is why we changed the initial Ang II model to CaPO4 model.

As suggested, we have analyzed the % of SMA in the CaPO4 model. Consistent with the result in Ang II model, CL316,243 administration significantly increased α -SMA content in the aortic wall in the CaPO4 model. We have added these results in the revised manuscript (Figure 6G-H).

Minor concerns

-The analysis of SMA in the figure 8 is performed in less animals than the analysis of elastin content. Please explain why.

Response: Thank you for the question. Due to the limited amount of pathological tissue from each mouse aneurysm, we used aortic sections from all mice (12 mice) for HE and EVG staining, and then we randomly used aortic sections from half of the mice (6 mice) for immunohistochemical staining and the other half (6 mice) for immunofluorescence staining. That is why we have less animal in the analysis of SMA in Figure 8G than the analysis of elastin content in Figure 8E.

-The references of the antibodies used should be included. In this respect, we guess the antibody used for caspase 3 (CST) is Cleaved Caspase-3 (Asp175) Antibody #9661 from Cell Signalling Technology, and then the authors should test total caspase 3 levels (#9662

from Cell Signalling Technology) .

Response: Thank you for the valuable suggestion. We have included all the detailed antibody information in the Reagents and Tools Table. In addition, we have detected total Caspase 3 levels and incorporated those results into the revised figures.

- There are some asterisks inside the figures (eg. Check figure 4F)

Response: Thank you for pointing this out. We have deleted those asterisks inside the figures.

-Line 49: "Several clinical studies...6-8." This statement includes experimental studies, please revise.

Response: Thank you for the very careful reading. We have corrected this statement as "Several clinical and animal studies" in the revised manuscript.

-Fig 2B. There should be some interesting downregulated proteins, so they should also be included.

Response: Thank you for the suggestion. We have analyzed the downregulated genes in aortic tissues from mice, and the data are shown below.

Figure. Gene ontology (GO) analysis of down-regulated genes in the aortic tissues from mice treated with saline or CL316,243 infused with Ang II.

-Fig S2.-Serum lipid analysis should be also analyzed in this model (<https://doi.org/10.1016/j.phrs.2021.105524>)

Response: Thank you for the suggestion. We have analyzed the serum triglyceride and cholesterol levels in our model. As shown below, 9-day CL316,243 administration significantly reduced serum triglyceride levels in ApoE^{-/-} mice but had no effect on cholesterol levels (data shown below). We have added these data in Fig EV2 (previously Fig S2).

Figure. Gene ontology (GO) analysis of down-regulated genes in the aortic tissues from mice treated with saline or CL316,243 infused with Ang II.

The previous study (<https://doi.org/10.1016/j.phrs.2021.105524>, PMID: 33667684) reported that long-term treatment of CL316,243 (3-12 weeks) reduced both triglyceride and cholesterol levels in APOE*3-Leiden.CETP mice (data shown below). At the same time, it can be inferred from their results that 1–2-week CL316,243 administration reduced plasma triglyceride levels but not cholesterol levels (data shown below), which is consistent with our results.

-The analysis of glucose were performed in fasting animals? Diabetes 1997 Aug;46(8):1257-63. doi: 10.2337/diab.46.8.1257.

Response: Thank you for the question. The analysis of glucose was performed in non-fasting animals. The previous study (Diabetes 1997 Aug;46(8):1257-63. doi: 10.2337/diab.46.8.1257, PMID: 9231648) showed that CL316,243 did not change both fed and fast blood glucose, which is consistent with our conclusion.

Referee #3 (Comments on Novelty/Model System for Author):

The manuscript "Brown remodeling of white adipose tissue protects against abdominal aortic aneurysm via a novel batokine FSTL1" (Chunling Huang et al, #EMM-2024-20948) describes the effect of browning of the white adipose tissue on abdominal aortic aneurysm (AAA) in two mouse models of AAA. The authors employed RNA sequencing and proteomic analysis to identify Follistatin-like 1 (FSTL1) secreted during browning as a

protective factor. The mechanism and signaling of this protection were addressed, and it was found that FSTL1 inhibited VSMC apoptosis through DIP2A/AKT. The effect of FSTL1 was confirmed in adipocyte-specific deficiency of the adipokine that resulted in a loss of the effect of browning on AAA (infection of Flox-FSTL1 mice with adenovirus carrying adipocyte-specific Cre). Supplementation of FSTL1 in AAA model, where there is a decrease of browning, protected from AAA. Importantly, the study was conducted in two mouse models and in cultured VSMC and includes human tissue samples of perivascular adipose tissue from AAA patients and control tissue from patients without AAA. The results are consistent across the models.

The manuscript is written in a clear logical manner and offers a clear representation of the results that are novel and assign a new vascular function to FSTL1. Evaluation of the effects is thorough and includes multiple browning markers, two AAA models, and several indexes of AAA and browning. The effect of systemic injection after AAA development is impressive and demonstrates a therapeutic potential.

Overall, this is a complete and well-designed and well-executed study with important conclusions and mechanistic insights. Lack of some important methodological details complicates the evaluation of the results quality. Methods section is incomplete and should be revised as suggested below, with detailed description of all approaches used to generate the results.

Response: Thank you for the highly positive comments, which greatly encourages us to improve our manuscript. As you suggested, we have added more methodological details in the revised methods.

1. I am not sure why FSTL1 is called "novel" - there is quite large body of literature on this protein, including publications describing effects in cardiovascular system, secretion by AT, and effects on VSMC. One report from 2012 shows that FSTL1 levels predict development of AAA (PMID: 22316625). Sadly, these publications have not been cited. The authors should revise the references to add all these publications, including a couple of recent reports from 2024.

Response: Sorry that we didn't describe it clear and thank you for the constructive suggestions. Yes, there are many literatures about the role of FSTL1 in cardiovascular system and metabolic tissues respectively. However, the role of FSTL1 as a batokine in vascular protection has not been reported, highlighting the novelty of this finding. We have carefully examined the manuscript and revised several inaccurate statements using "novel", including the title and the text, to improve clarity.

In the report from 2012, Mark Gorelik et al found that the plasma levels of FSTL1 were elevated in toddler patients with acute Kawasaki disease (KD), suggesting its level may be associated with acute arterial injury and coronary artery aneurysms (CAA) formation. Considering the relatively small numbers of patients in the CAA group (only 7 cases) and the difference between CAA and AAA, the relationship between FSTL1 level and AAA development remains unclear. In this study, we found that plasma levels of FSTL1 were significantly lower in AAA patients, suggesting that reduced levels of FSTL1 may be associated with AAA development. However, whether FSTL1 level could predict AAA development still needs further investigation.

We have added this discussion in the revised manuscript.

Besides, we have revised the references to complete the publications describing effects of FSTL1 in cardiovascular system, adipose tissue and VSMCs, including those published in 2024 and 2025.

2. Evaluation of browning should be described in Methods or in supplement, with references to original reports describing the markers.

Response: *Thank you for the suggestions. we have added a paragraph of description about the evaluation of browning in Methods, with references to original reports describing the markers.*

3. Quantification of staining should be described in Methods or Supplement.

Response: *Thank you for the suggestion. We have added description of the quantification of staining in Methods.*

4. Why were two mouse AAA models used? This should be explained and discussed.

Response: *Thank you for the question. Ang II model and CaPO4 model are two commonly used mouse models for studying AAA. While Ang II model mimics AAA driven by systemic factors such as hypertension, oxidative stress, and inflammation, the CaPO4 model replicates localized vascular injury, mimicking medial degeneration, elastin breakdown, and calcification in the infrarenal aorta. Therefore, findings confirmed in both models are more robust and translatable, minimizing model-specific bias. In addition, the Ang II infusion model results in the formation of aortic aneurysms at a variable location, whereas CaPO4 incubation model allows for the operation to be performed at a fixed location of the abdominal aorta, which facilitates hydrogel wrapping for our later study. That is why we used both Ang II model and CaPO4 mode in this manuscript.*

5. The selection of FST1 out of 18 proteins that were upregulated in Fig.4(A-D) has not been discussed or justified. Were other proteins also tested? Are they involved as well? There are several that are known to effect VSMC functions. These results should be also discussed in the Discussion section.

Response: *Thank you for your insightful comments and good suggestions. Given the critical role of VSMC apoptosis in AAA progression and our gene ontology (GO) analysis indicates browning adipocytes significantly inhibit VSMC apoptosis, FSTL1 emerges as a compelling candidate due to its well-documented anti-apoptotic effects across multiple cell types and its cardioprotective role. As the reviewer pointed out, our proteomic profiling also detected some other interesting proteins with established roles in VSMCs such as periostin and thrombospondin-4. Periostin is an extracellular matrix protein known to regulate VSMC migration and osteoblastic switch, and has been suggested to promote atherosclerosis and vascular calcification (PMID: 16325820, PMID: 33744420, PMID: 33834866, PMID: 36009051). Thrombospondin-4 is a member of the extracellular calcium-binding protein family and is linked to cell adhesion and migration (PMID: 26868511), and has been reported to*

promote VSMC proliferation, local inflammation, atherogenesis and restenosis (PMID: 12952849, PMID: 20884877, PMID: 26868511). Given our focus on identifying protective mediators of adipocyte-VSMC crosstalk, we prioritized targets with clearer therapeutic potential and thus excluded periostin and thrombospondin-4. It is worth noting that another two proteins, Serpin A9 and serpinA3N, both of which belong to the serine protease inhibitor (serpin) superfamily, are also detected. Given their anti-apoptotic functions in neurological contexts and tumor milieus (PMID: 34897996, PMID: 23892923, PMID: 17479112), investigation of their potential protective roles in AAA would be of significant interest in the future study. We have added this discussion in the revised manuscript.

6. In Figure 5C and D, it does not seem that representative blots were used: while the quantification shows no decrease or even increase of PARP in si-DIP2A-treated cells in response to FSTL1, there is an obvious decrease in the blots.

Response: Thank you for pointing this out. We have replaced Figure 5C and 5D with a more representative images in the revised manuscript.

7. Figure legends should be revised to include explanations of abbreviated labels in figures (including Supplemental Figures), even if abbreviations are spelled out in the text.

Response: Thank you for the suggestion. we have revised the figure legends to include explanations of abbreviated labels in figures and supplemental figures.

8. The discussion largely repeats the introduction - these sections should be reconciled. The discussion sections should be used to discuss the current study and results, implications and limitations etc.

Response: Thank you for the valuable suggestion. We have revised the discussion, delving deeper into the analysis of the results and implications. We also added implications in the revised discussion.

9. Evaluation of elastin fragmentation should be better described in Methods (Supplement). What were criteria for no elastin degradation, mild degradation, moderate, moderate to severe, severe elastin degradation? How was this quantified?

Response: Thank you for the suggestion. The criteria for different grades of elastin fragmentation are based on previous literatures (Circulation. 2020, PMID: 32354235; Circulation. 2023, PMID: 38018467) and are briefly as follows: 1, no elastin degradation; 2, mild degradation: degradation of less than 25%; 3, moderate

degradation: degradation of 25–50%; 4, moderate to severe degradation: degradation of 50–75% 5, severe degradation: degradation of more than 75%. We have added this description in the Methods.

Minor:

a) Please correct the second sentence in the abstract, "adipose tissue" used twice: "Emerging evidences have suggested a crosstalk between adipose tissue and vascular cells and brown adipose tissue is beneficial for cardiovascular health." "Evidence" should be in singular form, "emerging evidence".

Response: *Thank you for pointing this out. We have corrected the second sentence in the abstract in the revised manuscript.*

b) Spellcheck is recommended - there are typos, e.g., "decreased" is misspelled several times as "deceased", there are other typos in the text.

Response: *We feel sorry for these mistakes and thanks a lot for the careful readings. We have carefully checked the text and modified those mistakes in the revised manuscript.*

1st Jul 2025

Dear Prof. Liu,

Thank you for submitting your manuscript to EMBO Molecular Medicine. We have now received the reports from referees #2 and #3. As you will see below, while referee #3 is satisfied with the revisions, referee #2 still has some concerns that should be addressed for your manuscript to be considered further by EMM.

As EMBO Press usually only allows one round of revisions, please be aware that this will be your last opportunity to address these issues. The revised manuscript will be reviewed again, and we cannot guarantee a positive outcome at this stage.

Moreover, please address the following editorial requests:

1/ Manuscript text:

- Please indicate in track changes mode any new modification in the text.
- Please note that an email bounced for Boshui Huang (huangbsh5@mail.sysu.edu.cn), please check and correct accordingly.
- Please note that all corresponding authors are required to supply an ORCID ID for their name upon submission of a revised manuscript. ORCID identifiers are currently missing for J. Wang and Y. Chen.
- Please remove the reagents and tools table from the manuscript text and upload it as a separate file.
- Data availability: please note that all datasets must be deposited, and publicly available before acceptance of the manuscript. Please provide URLs linking to each dataset.
- Please provide a disclosure and competing interests statement. Please review the policy <https://www.embopress.org/competing-interests> and update your competing interests.
- Correct the reference formatting to alphabetical order and limit the number of author names listed to 10, followed by et al.
- Please add headings: Figure Legends, Expanded View Figure Legends

2/ Figures:

- Please make sure that all figures and figure panels are referenced in the text. Currently, a callout is missing for Fig. 6G, H.
- Figure re-use is allowed if justified, but must be mentioned in the legends (i.e. Figures 2C and G)
- Please rename Table EV1 to Table 1.
- Please address the queries from our data editors in the figure legends:
 1. Please note that the exact p values are not provided in the legends of figures 1B, C, D, M; 2D, F, H; 4G, H, J; 6H; 7C, H; 8G, I; EV1 C, EV2 C, EV5 A
 2. Please note that information related to n is missing in the legend of figure 4A
 3. Please note that the error bars are not defined in the legends of figures 1B, C, D, M, O; 2D, F, H; 3C, E, G, H; 4F-K; 5A, B, C, D, F, G, H, I; 5B, F; 6C, D, F, H; 7C, D, F, H; 8C, E, G, I, K; EV1 C, EV3 A, B; EV4 B, EV5 A, EV6 A, B.

3/ Please provide 'The paper explained': EMBO Molecular Medicine articles are accompanied by a summary of the articles to emphasize the major findings in the paper and their medical implications for the non-specialist reader. Please provide a draft summary of your article highlighting

4/ Thank you for providing a nice visual abstract. Please resize to file 550 px wide x 300-600 px high. A cropped portion of this image will serve as thumbnail for the table of content on our webpage.

Please also provide a synopsis text, that should include a short stand first (maximum of 300 characters, including space) as well as 2-5 one-sentences bullet points that summarizes the paper (maximum of 30 words / bullet point).

5/ As part of the EMBO Publications transparent editorial process initiative (see our Editorial at

<http://embomolmed.embopress.org/content/2/9/329>), EMBO Molecular Medicine will publish online a Review Process File (RPF) to accompany accepted manuscripts.

This file will be published in conjunction with your paper and will include the anonymous referee reports, your point-by-point response and all pertinent correspondence relating to the manuscript. Let us know whether you agree with the publication of the RPF and as here, if you want to remove or not any figures from it prior to publication.

I look forward to receiving your revised manuscript.

Yours sincerely,

Lise Roth

***** Reviewer's comments *****

Referee #2 (Comments on Novelty/Model System for Author):

The new results derived from the human plasma cohort has not been described in the methods so it is impossible to check the clinical characteristics, as well as ethical issues. Moreover, the corresponding statistical analysis is not completed (multivariate regression analysis has not been performed)

Referee #2 (Remarks for Author):

Although the authors have answered to most of the requested questions, there are still some issues that should be addressed:

- The authors have included a new larger set of controls and AAA patients but they do not mention how they obtained this cohort in the methods and they do not show the clinical characteristics. Moreover, multivariate logistic regression analysis should be performed to check for potential confounders.
- All the results presented in the response to reviewers should be discussed and shown (at least as supplemental figures)

Referee #3 (Remarks for Author):

The revision improved the manuscript. My concerns were adequately addressed in the revised version. I have no further suggestions.

RESPONSE TO EDITOR AND REVIEWERS

REVIEWER COMMENTS

Referee #2 (Comments on Novelty/Model System for Author):

The new results derived from the human plasma cohort has not been described in the methods so it is impossible to check the clinical characteristics, as well as ethical issues. Moreover, the corresponding statistical analysis is not completed (multivariate regression analysis has not been performed)

***Response:** Thank you for the valuable suggestions. We have added the description about the human plasma sample collection in the methods and provided the baseline clinical characteristics in Table 2 in the revised manuscript. Moreover, we have performed multivariable logistic regression analysis to address for potential confounders. The details will be shown below.*

Referee #2 (Remarks for Author):

Although the authors have answered to most of the requested questions, there are still some issues that should be addressed:

- The authors have included a new larger set of controls and AAA patients but they do not mention how the obtained this cohort in the methods and they do not show the clinical characteristics. Moreover, multivariate logistic regression analysis should be performed to check for potential confounders.

***Response:** Plasma samples were collected from patients diagnosed with AAA by abdominal ultrasound. Control participants who were matched to AAA cases by age, sex and date of hospitalization were selected. We have added this description in the methods. Besides, we have provided the clinical characteristics of AAA patients and controls in Table 2.*

Table 2. Baseline characteristics of patients with and without AAA for serum sampling

Variables	Without AAA (n=68)	With AAA (n=55)	P value
Men (%)	61 (89.7%)	49 (87.5%)	0.919
Age (years)	69.4 (5.46)	70.3 (5.18)	0.346
Smoker (%)	31 (45.6%)	26 (46.4%)	1.000
Drinker (%)	14 (20.6%)	15 (26.8%)	0.550
Body mass index, kg/m ²	23.7 (4.60)	24.0 (2.85)	0.608
Systolic blood pressure, mmHg	128 (21.0)	138 (21.0)	0.008
Diastolic blood pressure, mmHg	76.4 (12.0)	78.4 (10.7)	0.318
LDL, mmol/L	2.72 (0.93)	2.81 (1.17)	0.649
HDL, mmol/L	1.07 (0.36)	0.99 (0.32)	0.164
Heart rate, bpm (%)	80.0 (16.4)	76.8 (13.2)	0.236
Hypertension (%)	33 (48.5%)	48 (85.7%)	<0.001
Diabetes mellitus (%)	20 (29.4%)	21 (37.5%)	0.447
FSTL1, ng/mL	41.3 (21.7)	26.1 (11.2)	<0.001

Note: AAA, abdominal aortic aneurysm

In addition, we have performed multivariable logistic regression analysis to adjust for all covariates presented at Table 2. The results indicated that serum FSTL1 levels remained significantly associated with AAA after adjustment (OR 0.94, 95% CI 0.90-0.96, $p < 0.001$), suggesting a potential role of FSTL1 in AAA development.

- All the results presented in the response to reviewers should be discussed and shown (at least as supplemental figures)

***Response:** Thank you for the good suggestion. We have put those results into supplemental figures (Fig EV4, EV8 and EV9) and added some discussion about those results in the revived manuscript.*

Referee #3 (Remarks for Author):

The revision improved the manuscript. My concerns were adequately addressed in the revised version. I have no further suggestions.

***Response:** Thank you very much for dedicating your valuable time to review our manuscript.*

EDITOR COMMENTS

1/ Manuscript text:

- Please indicate in track changes mode any new modification in the text.

***Response:** We have made the modification in track change mode in the text.*

- Please note that an email bounced for Boshui Huang (huangbsh5@mail.sysu.edu.cn), please check and correct accordingly.

***Response:** We have double-checked Boshui Huang's email address (huangbsh5@mail.sysu.edu.cn) and confirmed it is correct*

- Please note that all corresponding authors are required to supply an ORCID ID for their name upon submission of a revised manuscript. ORCID identifiers are currently missing for J. Wang and Y. Chen.

***Response:** The ORCID for J. Wang and Y. Chen is 0000-0002-5827-7876 and 0000-0003-2051-9320, respectively.*

- Please remove the reagents and tools table from the manuscript text and upload it as a separate file.

***Response:** The reagents and tools table has been removed from the manuscript and uploaded as a separate file.*

- Data availability: please note that all datasets must be deposited, and publicly available before acceptance of the manuscript. Please provide URLs linking to each dataset.

***Response:** The URLs linking to each dataset has been provided in the revised manuscript.*

- Please provide a disclosure and competing interests statement. Please review the

policy <https://www.embopress.org/competing-interests> and update your competing interests.

Response: *Disclosure and competing interests statement has been provided in the revised manuscript.*

- Correct the reference formatting to alphabetical order and limit the number of author names listed to 10, followed by et al.

Response: *The reference formatting has been corrected.*

- Please add headings: Figure Legends, Expanded View Figure Legends

Response: *Headings have been added.*

2/ Figures:

- Please make sure that all figures and figure panels are referenced in the text. Currently, a callout is missing for Fig. 6G, H.

Response: *The missing information has been added.*

- Figure re-use is allowed if justified, but must be mentioned in the legends (i.e. Figures 2C and G)

Response: *To ensure clarity, we have replaced Figure 2C with new ones.*

- Please rename Table EV1 to Table 1.

Response: *Corrected as suggested.*

- Please address the queries from our data editors in the figure legends:

1. Please note that the exact p values are not provided in the legends of figures 1B, C, D, M; 2D, F, H; 4G, H, J; 6H; 7C, H; 8G, I; EV1 C, EV2 C, EV5 A

Response: *Corrected as suggested.*

2. Please note that information related to n is missing in the legend of figure 4A

Response: *n=3 in figure 4A. We have added this information in the legend.*

3. Please note that the error bars are not defined in the legends of figures 1B, C, D, M, O; 2D, F, H; 3C, E, G, H; 4F-K; 5A, B, C, D, F, G, H, I; 5B, F; 6C, D, F, H; 7C, D, F, H; 8C, E, G, I, K; EV1 C, EV3 A, B; EV4 B, EV5 A, EV6 A, B.

Response: *Data are presented as the means \pm SEM. This information has been provided in the Statistical analysis in the Methods.*

3/ Please provide 'The paper explained': EMBO Molecular Medicine articles are accompanied by a summary of the articles to emphasize the major findings in the paper and their medical implications for the non-specialist reader. Please provide a draft summary of your article highlighting

- the medical issue you are addressing,
- the results obtained and

- their clinical impact.

Response: We have added this summary part as follows:

The paper explained:

Problem: The impact of adipose tissue on abdominal aortic aneurysm (AAA) development has received much attention, but whether brown remodeling of white adipose tissue would protect against AAA remains unclear.

Results: Patients with AAA had a decreased browning level of adipose tissue and browning induction in mice attenuated AAA development. Browning adipocytes secrete a novel vessel-protective adipokine Follistatin-like 1 (FSTL1) to inhibit vascular smooth muscle cell apoptosis. Adipocyte-specific deficiency of FSTL1 abrogated the protective effect of browning induction and supplementation of FSTL1 inhibited AAA development.

Impact: Our study suggests the therapeutic potential of adipose tissue browning and FSTL1 supplementation for treating AAA.

4/ Thank you for providing a nice visual abstract. Please resize to file 550 px wide x 300-600 px high. A cropped portion of this image will serve as thumbnail for the table of content on our webpage. Please also provide a synopsis text, that should include a short stand first (maximum of 300 characters, including space) as well as 2-5 one-sentences bullet points that summarizes the paper (maximum of 30 words / bullet point).

Response: We have resized the file to 550 px wide x 300-600 px high and provided a synopsis text as suggested.

Synopsis:

Brown remodeling of white adipose tissue was found to inhibit abdominal aortic aneurysm (AAA) progression via a vessel-protective adipokine Follistatin-like 1 (FSTL1), suggesting a novel therapeutic strategy for AAA intervention.

- Browning level is decreased in AAA patients and browning induction in mice attenuates AAA development.

- *Browning adipocytes secret FSTL1 and inhibits VSMC apoptosis via receptor Dip2A-mediated Akt activation.*
- *Adipocyte-specific deficiency of FSTL1 abrogates the protective effect of browning induction.*
- *Supplementation of recombinant FSTL1 attenuates AAA development.*

5/ As part of the EMBO Publications transparent editorial process initiative (see our Editorial at <http://embomolmed.embopress.org/content/2/9/329>), EMBO Molecular Medicine will publish online a Review Process File (RPF) to accompany accepted manuscripts.

This file will be published in conjunction with your paper and will include the anonymous referee reports, your point-by-point response and all pertinent correspondence relating to the manuscript. Let us know whether you agree with the publication of the RPF and as here, if you want to remove or not any figures from it prior to publication.

Response: We agree with the publication of the RPF.

3rd Sep 2025

Dear Prof. Liu,

Thank you for submitting your revised study, and please accept my apologies for the delay in getting back to you during this busy time of the year. We have now received the report from referee #2, who is overall satisfied with the revisions. I will therefore be able to accept your manuscript once the following minor concerns are addressed:

1/ Please carefully address the remaining concerns raised by the referee.

2/ Please note that all corresponding authors are required to supply an ORCID ID for their name upon submission of a revised manuscript. ORCID identifiers are still missing for J. Wang and Y. Chen.

3/ Manuscript text:

- Please accept previous changes and only keep in track changes mode any new modification in the text.

- Methods:

o Human samples: as also requested by referee #2, please clarify whether the statement on informed consent and Helsinki declaration applies to plasma samples. Please also state details of authority granting ethics approval and provide reference number for approval for plasma samples. If collected and within the bounds of privacy constraints report on age, sex and gender or ethnicity for all study participants.

o Cells: please indicate whether the cells were tested for mycoplasma contamination.

o Antibodies: please provide dilutions/concentrations

o Statistics: please provide a statement on sample size, inclusion/exclusion criteria, blinding and randomization.

- Data availability section: please remove "All data supporting the findings of this study are available in the article file and Supplemental material file, or available from the authors upon reasonable request." Please note that deposited datasets must be publicly available before acceptance of the manuscript. Please provide updated URLs for these datasets.

4/ Thank you for providing Source Data. Please also provide source data for Figure 1E to 1H.

5/ Checklist:

- Cell materials: please indicate whether the cells were tested for mycoplasma contamination.

- Human research participants: please check whether Table EV1 was provided.

- Laboratory protocol: please check that this subsection applies to your study.

- Experimental study design and statistics: please fill in the inclusion/exclusion criteria subsection.

6/ I added minor edits to you Paper Explained. Please let me know if you agree or amend as you see fit, and include the text in the main manuscript file:

Problem: The impact of adipose tissue on abdominal aortic aneurysm (AAA) development has received much attention, but whether brown remodeling of white adipose tissue would protect against AAA remains unclear.

Results: Patients with AAA had a decreased browning level of adipose tissue and browning induction in mice attenuated AAA development. Browning adipocytes secreted a vessel-protective adipokine, Follistatin-like 1 (FSTL1), that inhibited vascular smooth muscle cell apoptosis. Adipocyte-specific FSTL1 deficiency abrogated the protective effect of browning induction, and supplementation of FSTL1 inhibited AAA development.

Impact: Our study suggests the therapeutic potential of adipose tissue browning and FSTL1 supplementation for treating AAA.

7/ Thank you for providing a nice visual abstract. I have cropped a small portion to serve as a thumbnail on the table of content for our webpage. Please let me know if you agree, or provide another image at the same dimensions.

I have added minor edits to your text, please let me know if you agree or amend as you see fit:

"Brown remodeling of white adipose tissue inhibited abdominal aortic aneurysm (AAA) progression via a vessel-protective adipokine Follistatin-like 1 (FSTL1), suggesting a novel therapeutic strategy for AAA intervention.

- Browning level is decreased in AAA patients and browning induction in mice attenuates AAA development.
- Browning adipocytes secrete FSTL1 and inhibit VSMC apoptosis via receptor Dip2A-mediated Akt activation.
- Adipocyte-specific FSTL1 deficiency abrogates the protective effect of browning induction.
- Supplementation of recombinant FSTL1 attenuates AAA development."

We note that you agree with the publication of the Review Process File.

I look forward to receiving your revised manuscript.

Yours sincerely,

Lise Roth

Lise Roth, PhD

Senior Editor

EMBO Molecular Medicine

***** Reviewer's comments *****

Referee #2 (Comments on Novelty/Model System for Author):

The description of statistical analysis is not completed. There is no information regarding ethical issues in patients and controls (plasma sample study)

Referee #2 (Remarks for Author):

The authors have included a table with the clinical characteristics but they should include additional data (how controls were chosen to exclude presence of AAA, and how matching was performed, as well as ethical issues). The detailed statistical analysis must be included in the methods section

RESPONSE TO EDITOR AND REVIEWERS**REVIEWER COMMENTS**

Referee #2 (Comments on Novelty/Model System for Author):

The description of statistical analysis is not completed. There is no information regarding ethical issues in patients and controls (plasma sample study)

***Response:** Thank you for the valuable suggestions. We have revised the Methods section to provide a more complete description of the statistical analysis and to clarify the ethical considerations for the plasma samples. Specifically, continuous and categorical variables were analyzed with appropriate parametric or non-parametric tests based on distributional assumptions, and multivariable logistic regression was performed adjusting for age, sex, smoking status, drinking status, body mass index, systolic and diastolic blood pressure, heart rate, LDL, HDL, hypertension, and diabetes. Results were reported as odds ratios with 95% confidence intervals. The collection of human plasma samples was conducted in accordance with the approved protocol by the Sun Yat-sen Memorial Hospital Ethics Committee (SYSKY-2023-1135-01). Informed consent was obtained from all participants and the experiments conformed to the principles set out in the WMA Declaration of Helsinki and the Department of Health and Human Services Belmont Report.*

Referee #2 (Remarks for Author):

The authors have included a table with the clinical characteristics but they should include additional data (how controls were chosen to exclude presence of AAA, and how matching was performed, as well as ethical issues). The detailed statistical analysis must be included in the methods section

***Response:** Thank you for the valuable suggestions. We have revised the Methods section to provide additional details as requested. Specifically, control participants were confirmed to be free of AAA by abdominal ultrasound and were matched to AAA patients by age (± 2 years), sex, and date of hospitalization. We have also clarified the ethical procedures, noting that both plasma and tissue samples were collected under approval from the Sun Yat-sen Memorial Hospital Ethics Committee (approval number SYSKY-2023-1135-01), with written informed consent obtained from all participants in accordance with the Declaration of Helsinki and the Belmont Report. In addition, we have included detailed statistical analysis in the Methods section.*

16th Sep 2025

Dear Prof. Liu,

Thank you for bearing with the last editorial requests. I am pleased to inform you that your manuscript is accepted for publication and is now being sent to our publisher to be included in the next available issue of EMBO Molecular Medicine!

Yours sincerely,

Lise Roth
